# Parallel-mentoring for Offline Model-based Optimization

**Can (Sam) Chen[1,2]\*, Christopher Beckham[2,3], Zixuan Liu[5], Xue Liu[1,2], Christopher Pal[2,3,4]**

[1]McGill University, [2]MILA - Quebec AI Institute,
[3]Polytechnique Montreal, [4]Canada CIFAR AI Chair, [5]University of Washington

## Abstract

We study offline model-based optimization to maximize a black-box objective function with a static dataset of designs and scores. These designs encompass a variety of domains, including materials, robots and DNA sequences. A common approach trains a proxy on the static dataset to approximate the black-box objective function and performs gradient ascent to obtain new designs. However, this often results in poor designs due to the proxy inaccuracies for out-of-distribution designs. Recent studies indicate that: (a) gradient ascent with a mean ensemble of proxies generally outperforms simple gradient ascent, and (b) a trained proxy provides weak ranking supervision signals for design selection. Motivated by (a) and (b), we propose *parallel-mentoring* as an effective and novel method that facilitates mentoring among parallel proxies, creating a more robust ensemble to mitigate the out-of-distribution issue. We focus on the three-proxy case and our method consists of two modules. The first module, *voting-based pairwise supervision*, operates on three parallel proxies and captures their ranking supervision signals as pairwise comparison labels. These labels are combined through majority voting to generate consensus labels, which incorporate ranking supervision signals from all proxies and enable mutual mentoring. However, label noise arises due to possible incorrect consensus. To alleviate this, we introduce an *adaptive soft-labeling* module with soft-labels initialized as consensus labels. Based on bi-level optimization, this module fine-tunes proxies in the inner level and learns more accurate labels in the outer level to adaptively mentor proxies, resulting in a more robust ensemble. Experiments validate the effectiveness of our method. Our code is available here.

## 1 Introduction

Designing new objects or entities to optimize specific properties is a widespread challenge, encompassing various domains such as materials, robots, and DNA sequences [1]. Traditional approaches often involve interacting with a black-box function to propose new designs, but this can be expensive or even dangerous in some cases [2–6]. In response, recent work [1] has focused on a more realistic setting known as offline model-based optimization (MBO). In this setting, the objective is to maximize a black-box function using only a static (offline) dataset of designs and scores.

A prevalent approach to addressing the problem is to train a deep neural network (DNN) model parameterized as $f_{\theta}(\cdot)$, on the static dataset, with the trained DNN serving as a proxy. The proxy allows for gradient ascent on existing designs, generating improved designs by leveraging the gradient information provided by the DNN model. However, this method encounters an issue with the trained proxy being susceptible to out-of-distribution problems. Specifically, the proxy produces inaccurate predictions when applied to data points that deviate significantly from the training distribution.

---

\*Correspondence to can.chen@mila.quebec.

37th Conference on Neural Information Processing Systems (NeurIPS 2023).

Recent studies have observed that (a) employing a mean ensemble of trained proxies for gradient ascent in offline MBO generally leads to superior designs compared to using a single proxy [7]. This improvement stems from the ability of the ensemble to provide more robust predictions compared to a single proxy [8–11]. Recent work has also found that (b) a trained proxy offers weak (valuable, albeit potentially unreliable) ranking supervision signals for design selection in various offline MBO contexts, such as evolutionary algorithms [12], reinforcement learning [13], and generative modeling [14]. These signals, focusing on the relative order of designs over absolute scores, are more resilient to noise and inaccuracies. By exchanging these signals among proxies in the ensemble, we can potentially enhance its robustness. As shown in Figure 1, we have three parallel proxies $f_{\theta}^A(\cdot)$, $f_{\theta}^B(\cdot)$ and $f_{\theta}^C(\cdot)$. For two designs

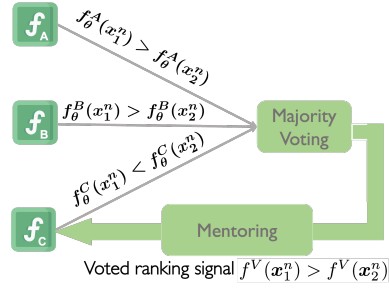

Figure 1: Motivation illustration.

$x_1^n$ and $x_2^n$ within the neighborhood of the current optimization point, proxies $f_{\theta}^A(\cdot)$ and $f_{\theta}^B(\cdot)$ agree that the score of $x_1^n$ is larger than that of $x_2^n$, while proxy $f_{\theta}^C(\cdot)$ disagrees. Based on the majority voting principle, proxies $f_{\theta}^A(\cdot)$ and $f_{\theta}^B(\cdot)$ provide a more reliable ranking, and their voted ranking signal $f^V(x_1^n) > f^V(x_2^n)$ could mentor the proxy $f_{\theta}^C(\cdot)$, thus enhancing its performance.

To this end, we propose an effective and novel method called *parallel-mentoring* that facilitates mentoring among parallel proxies to train a more robust ensemble against the out-of-distribution issue. This paper primarily focuses on the three-proxy case, referred to as *tri-mentoring*, but we also examine the situation with more proxies in Appendix A.1. As depicted in Figure 2, *tri-mentoring* consists of two modules. **Module 1**, *voting-based pairwise supervision* (shown in Figure 2(a)), operates on three parallel proxies $f_{\theta}^A(\cdot)$, $f_{\theta}^B(\cdot)$, and $f_{\theta}^C(\cdot)$ and utilizes their mean for the final prediction. To ensure consistency with the ranking information employed in design selection, this module adopts a pairwise approach to represent the ranking signals of each proxy. Specifically, as illustrated in Figure 2(a), this module generates samples (e.g. $x_1^n$, $x_2^n$ and $x_3^n$) in the neighborhood of the current point $x$ and computes pairwise comparison labels $\hat{y}^A$ for all sample pairs, serving as ranking supervision signals for the proxy $f_{\theta}^A(\cdot)$. The label $\hat{y}_{ij}^A$ is defined as 1 if $f_{\theta}^A(x_i) > f_{\theta}^A(x_j)$ and 0 otherwise, and similar signals are derived for proxies $f_{\theta}^B(\cdot)$ and $f_{\theta}^C(\cdot)$. These labels $\hat{y}^A$, $\hat{y}^B$ and $\hat{y}^C$ are combined via majority voting to create consensus labels $\hat{y}^V$ which are more reliable and thus can be used for mentoring proxies. The voted ranking signal $f^V(x_1^n) > f^V(x_2^n)$ in Figure 1 corresponds to the pairwise consensus label $\hat{y}_{12}^V = 1$ in Figure 2(a), and both can mentor the proxy $f_{\theta}^C(\cdot)$.

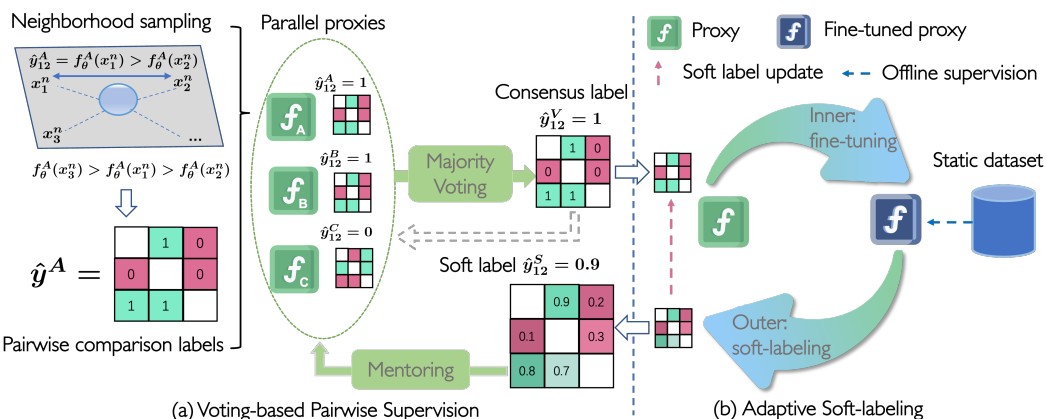

Figure 2: Illustration of *tri-mentoring*.

**Module 2**, *adaptive soft-labeling* (shown in Figure 2(b)), mitigates the issue of label noise that may arise, since the voting consensus may not always be correct. To this end, this module initializes the consensus labels $\hat{y}^V$ from the first module as soft-labels $\hat{y}^S$. It then aims to learn more accurate soft-labels to better represent the ranking supervision signals by leveraging the knowledge from the static dataset. Specifically, assuming accurate soft-labels, one of the proxies, either $f_{\theta}^A(\cdot)$, $f_{\theta}^B(\cdot)$ or $f_{\theta}^C(\cdot)$, fine-tuned using them, is expected to perform well on the static dataset, as both soft-labels

(pairwise perspective) and the static dataset (pointwise perspective) describe the same ground-truth and share underlying similarities. This formulation leads to a bi-level optimization framework with an inner *fine-tuning* level and an outer *soft-labeling* level as shown in Figure 2(b). The inner level fine-tunes the proxy with soft-labels, which establishes the connection between them. The outer level optimizes soft-labels to be more accurate by minimizing the loss of the static dataset via the inner-level connection. The optimized labels are further fed back to the first module to adaptively mentor the proxy, ultimately yielding a more robust ensemble. Experiments on design-bench validate the effectiveness of our method.

To summarize, our contributions are three-fold:

- We propose *parallel-mentoring* for offline MBO, effectively utilizing weak ranking supervision signals among proxies, with a particular focus on the three-proxy case as *tri-mentoring*.
- Our method consists of two modules: *voting-based pairwise supervision* and *adaptive soft-labeling*. The first module generates pairwise consensus labels via majority voting to mentor the proxies.
- To mitigate label noise in consensus labels, the second module proposes a bi-level optimization framework to adaptively fine-tune proxies and soft-labels, resulting in a more robust ensemble.

## 2 Preliminaries: Gradient Ascent on Offline Model-based Optimization

Offline model-based optimization (MBO) aims to find the optimal design $\boldsymbol{x}^*$ that maximizes the black-box objective function $f(\cdot)$:

$$\boldsymbol{x}^* = \arg\max_{\boldsymbol{x}} f(\boldsymbol{x}), \tag{1}$$

To achieve this, a static dataset $\mathcal{D} = \{(\boldsymbol{x}_i, y_i)\}_{i=1}^{N}$ with $N$ points is available, where $\boldsymbol{x}_i$ represents a design and $y_i$ is its corresponding score.

A common approach for solving this optimization problem is to fit a deep neural network (DNN) model $f_{\boldsymbol{\theta}}(\cdot)$ with parameters $\boldsymbol{\theta}$ to the static dataset in a supervised manner. The optimal parameters $\boldsymbol{\theta}^*$ can be obtained by minimizing the mean squared error between the predictions and the true scores:

$$\boldsymbol{\theta}^* = \arg\min_{\boldsymbol{\theta}} \frac{1}{N} \sum_{i=1}^{N} (f_{\boldsymbol{\theta}}(\boldsymbol{x}_i) - y_i)^2. \tag{2}$$

The trained DNN model $f_{\boldsymbol{\theta}^*}(\cdot)$ acts as a proxy to optimize the design using gradient ascent steps:

$$\boldsymbol{x}_{t+1} = \boldsymbol{x}_t + \eta \nabla_{\boldsymbol{x}} f_{\boldsymbol{\theta}}(\boldsymbol{x})|_{\boldsymbol{x}=\boldsymbol{x}_t}, \quad \text{for } t \in [0, T-1], \tag{3}$$

where $T$ is the number of steps and $\eta$ represents the learning rate. $\boldsymbol{x}_T$ serves as the final design candidate. However, this method faces a challenge with the proxy being vulnerable to out-of-distribution designs. When handling designs that substantially differ from the training distribution, the proxy yields inaccurate predictions.

## 3 Method

In this section, we introduce *parallel-mentoring*, focusing on the three-proxy scenario, also known as *tri-mentoring*. The method can be easily extended to incorporate more proxies, as discussed in Appendix A.1. *Tri-mentoring* consists of two modules. The first module, *voting-based pairwise supervision* in Section 3.1, manages three proxies parallelly and generates consensus labels via majority voting to mentor proxies. To mitigate label noise, we introduce a second module, *adaptive soft-labeling* in Section 3.2. This module adaptively fine-tunes proxies and soft-labels using bi-level optimization, improving ensemble robustness. The overall algorithm is shown in Algorithm 1.

### 3.1 Voting-based Pairwise Supervision

We train three parallel proxies $f_{\boldsymbol{\theta}}^A(\cdot)$, $f_{\boldsymbol{\theta}}^B(\cdot)$ and $f_{\boldsymbol{\theta}}^C(\cdot)$ on the static dataset with different initializations and utilize their mean as the final prediction as suggested in [1, 15]:

$$f_{\boldsymbol{\theta}}(\cdot) = \frac{1}{3}(f_{\boldsymbol{\theta}}^A(\cdot) + f_{\boldsymbol{\theta}}^B(\cdot) + f_{\boldsymbol{\theta}}^C(\cdot)). \tag{4}$$

We then apply gradient ascent with $f_{\boldsymbol{\theta}}(\cdot)$ on existing designs to generate improved designs as per Eq.(3). Although the mean ensemble generally results in superior designs compared to a single proxy

**Algorithm 1 Tri-mentoring for Offline Model-based Optimization**

---

**Input:** The static dataset $\mathcal{D}$, the number of iterations $T$, the optimizer $OPT(\cdot)$.
**Output:** The high-scoring design $\boldsymbol{x}_h^*$.

1   Initialize $\boldsymbol{x}_0$ as the design with the highest score in $\mathcal{D}$.
2   Train proxies $f_{\boldsymbol{\theta}}^A(\cdot)$, $f_{\boldsymbol{\theta}}^B(\cdot)$ and $f_{\boldsymbol{\theta}}^C(\cdot)$ on $\mathcal{D}$ with different initializations.
3   **for** $t \leftarrow 0$ **to** $T-1$ **do**
       ▷ **Voting-based pairwise supervision.**
4       Sample $K$ neighborhood points at $\boldsymbol{x}_t$ as $\mathcal{S}(\boldsymbol{x}_t)$.
5       Compute pairwise comparison labels $\hat{\boldsymbol{y}}^A$, $\hat{\boldsymbol{y}}^B$ and $\hat{\boldsymbol{y}}^C$ for three proxies on $\mathcal{S}(\boldsymbol{x}_t)$.
6       Derive consensus labels: $\hat{\boldsymbol{y}}^V = \text{majority\_voting}(\hat{\boldsymbol{y}}^A, \hat{\boldsymbol{y}}^B, \hat{\boldsymbol{y}}^C)$.
       ▷ **Adaptive soft-labeling.**
7       **for** proxy **in** $[f_{\boldsymbol{\theta}}^A(\cdot), f_{\boldsymbol{\theta}}^B(\cdot), f_{\boldsymbol{\theta}}^C(\cdot)]$ **do**
8           Initialize soft-labels as consensus labels: $\hat{\boldsymbol{y}}^S = \hat{\boldsymbol{y}}^V$.
9           Inner level: fine-tune the proxy with Eq.(8).
10         Outer level: learn more accurate soft-labels $\hat{\boldsymbol{y}}^S$ with Eq.(9).
11         Mentor proxy using the optimized soft-labels $\hat{\boldsymbol{y}}^S$ with Eq.(8).
       ▷ **Gradient ascent with a mean ensemble.**
12      Form a more robust ensemble as $f_{\boldsymbol{\theta}}(\boldsymbol{x}) = \frac{1}{3}(f_{\boldsymbol{\theta}}^A(\boldsymbol{x}) + f_{\boldsymbol{\theta}}^B(\boldsymbol{x}) + f_{\boldsymbol{\theta}}^C(\boldsymbol{x}))$
13      Gradient ascent: $\boldsymbol{x}_{t+1} = \boldsymbol{x}_t + \eta OPT(\nabla_{\boldsymbol{x}} f_{\boldsymbol{\theta}}(\boldsymbol{x}_t))$
14   Return $\boldsymbol{x}_h^* = \boldsymbol{x}_T$

---

[7] due to the ensemble robustness [8, 9], this approach does not fully exploit the potential of weak (valuable, albeit potentially unreliable) ranking supervision signals within every proxy. Emphasizing the relative order of designs rather than their absolute scores, these signals are more resilient to noise and inaccuracies. These ranking signals are commonly used in evolutionary algorithms [12], reinforcement learning [13], and generative modeling [14] to select designs and could further improve the ensemble robustness. We extract the ranking supervision signals from individual proxies in the form of pairwise comparison labels, and then combine these labels via majority voting to generate consensus labels to mentor proxies. We provide a detailed explanation of this module below, with its implementation shown in Algorithm 1 from Line 4 to Line 6.

**Pairwise comparison label.** We adopt a pairwise approach to represent the ranking supervision signals for every proxy, focusing on relative order to align with the ranking information used in design selection. We sample $K$ points at the neighborhood of the optimization point $\boldsymbol{x}_t$ as $\mathcal{S}(\boldsymbol{x}_t)$ $= \{\boldsymbol{x}_1^n, \ldots, \boldsymbol{x}_K^n\} \sim \mathcal{N}(\boldsymbol{x}_t, \delta^2)$ where $\mathcal{N}(\boldsymbol{x}_t, \delta^2)$ represents a Gaussian distribution centered at $\boldsymbol{x}_t$ with variance $\delta^2$. For each sample pair $(\boldsymbol{x}_i^n, \boldsymbol{x}_j^n)$ and a proxy (e.g., $f_{\boldsymbol{\theta}}^A(\cdot)$), we define the pairwise comparison label $\boldsymbol{y}_{ij}^A = \mathbf{1}(f_{\boldsymbol{\theta}}^A(\boldsymbol{x}_i^n) > f_{\boldsymbol{\theta}}^A(\boldsymbol{x}_j^n))$, where $\mathbf{1}$ is the indicator function. The labels $\hat{\boldsymbol{y}}^A$ from all sample pairs serves as the ranking supervision signals for the proxy $f_{\boldsymbol{\theta}}^A(\cdot)$. We repeat this process for all proxies, generating signals $\hat{\boldsymbol{y}}^B$ and $\hat{\boldsymbol{y}}^C$ for proxies $f_{\boldsymbol{\theta}}^B(\cdot)$ and $f_{\boldsymbol{\theta}}^C(\cdot)$ respectively.

**Majority voting.** Given these pairwise comparison labels $\hat{\boldsymbol{y}}^A$, $\hat{\boldsymbol{y}}^B$ and $\hat{\boldsymbol{y}}^C$, we derive the pairwise consensus labels $\hat{\boldsymbol{y}}^V$ via an element-wise majority voting:

$$\hat{\boldsymbol{y}}_{ij}^V = \text{majority\_voting}(\hat{\boldsymbol{y}}_{ij}^A, \hat{\boldsymbol{y}}_{ij}^B, \hat{\boldsymbol{y}}_{ij}^C)\,, \tag{5}$$

where $i$ and $j$ are the indexes of the neighborhood samples. As consensus labels are generally more reliable, they can be employed for mentoring the proxies to promote the exchange of ranking supervision signals. Specifically, we can fine-tune the proxy $f_{\boldsymbol{\theta}}^A(\cdot)$ using the binary cross-entropy loss, where $\sigma(f_{\boldsymbol{\theta}}^A(\boldsymbol{x}_i^n) - f_{\boldsymbol{\theta}}^A(\boldsymbol{x}_j^n))$ represents the predicted probability that $f_{\boldsymbol{\theta}}^A(\boldsymbol{x}_i^n) > f_{\boldsymbol{\theta}}^A(\boldsymbol{x}_j^n)$, as also used in the ChatGPT reward model training [16–18]. The loss function can be computed as:

$$\mathcal{L}^A(\boldsymbol{\theta}) = -\frac{1}{C_K^2} \sum_{1 \leq i < j \leq K} \hat{\boldsymbol{y}}_{ij}^V \log[\sigma(f_{\boldsymbol{\theta}}^A(\boldsymbol{x}_i^n) - f_{\boldsymbol{\theta}}^A(\boldsymbol{x}_j^n))] + (1 - \hat{\boldsymbol{y}}_{ij}^V) \log[\sigma(f_{\boldsymbol{\theta}}^A(\boldsymbol{x}_j^n) - f_{\boldsymbol{\theta}}^A(\boldsymbol{x}_i^n))], \quad (6)$$

where $C_K^2 = \frac{K(K-1)}{2}$ denotes the number of the sample pairs. This procedure is also applied to proxies $f_{\boldsymbol{\theta}}^B(\cdot)$ and $f_{\boldsymbol{\theta}}^C(\cdot)$. While our approach encourages alignment with the consensus, it does not aim to make proxies identical. Each proxy maintains its unique learning trajectory, thereby preserving the diversity among the proxies.

## 3.2 Adaptive Soft-labeling

The consensus labels $\hat{\boldsymbol{y}}^V$ may contain noise since the majority voting consensus can be incorrect. To mitigate this issue, we propose an *adaptive soft-labeling* module. This module initializes soft-labels as consensus labels, and employs a bi-level optimization framework for adaptive mentoring, where the inner level fine-tunes the proxies and the outer level refines the soft-labels. We provide a detailed description of this module below; its implementation is outlined in Algorithm 1, Line 7- Line 11.

**Fine-tuning.** We initialize the soft-labels as the consensus labels: $\hat{\boldsymbol{y}}^S = \hat{\boldsymbol{y}}^V$, serving as an effective starting point. Utilizing the soft-labels $\hat{\boldsymbol{y}}^S$, we can perform fine-tuning on the proxy $f_{\boldsymbol{\theta}}^A(\cdot)$ against the binary cross-entropy loss following Eq.(6),

$$\mathcal{L}^A(\boldsymbol{\theta}, \boldsymbol{y}^S) = -\frac{1}{C_K^2} \sum_{1 \leq i < j \leq K} \hat{\boldsymbol{y}}_{ij}^S \log[\sigma(f_{\boldsymbol{\theta}}^A(\boldsymbol{x}_i) - f_{\boldsymbol{\theta}}^A(\boldsymbol{x}_j))] + (1 - \hat{\boldsymbol{y}}_{ij}^S) \log[\sigma(f_{\boldsymbol{\theta}}^A(\boldsymbol{x}_j) - f_{\boldsymbol{\theta}}^A(\boldsymbol{x}_i))] . \quad (7)$$

In contrast to Eq.(6), we express the loss $\mathcal{L}^A(\boldsymbol{\theta}, \hat{\boldsymbol{y}}^S)$ as a function of both the proxy parameters $\boldsymbol{\theta}$ and the soft-labels $\hat{\boldsymbol{y}}^S$ as the accurate soft-labels $\hat{\boldsymbol{y}}^S$ have yet to be determined. One-step gradient descent enables fine-tuning, resulting in the following relationship:

$$\boldsymbol{\theta}(\hat{\boldsymbol{y}}^S) = \boldsymbol{\theta} - \gamma \frac{\partial \mathcal{L}^A(\boldsymbol{\theta}, \hat{\boldsymbol{y}}^S)}{\partial \boldsymbol{\theta}^\top}, \quad (8)$$

where $\gamma$ denotes the fine-tuning learning rate.

**Soft-labeling.** Assuming the soft-labels are accurate, the fine-tuned proxy $f_{\boldsymbol{\theta}(\hat{\boldsymbol{y}}^S)}^A(\cdot)$ is expected to perform well on the static dataset. This is due to the fact that, despite having different data distributions, the soft-labels and the static dataset share underlying similarities, as they both represent the same ground-truth from pairwise and pointwise perspectives, respectively. This leads to a bi-level optimization framework with the fine-tuning mentioned above as the inner level and the soft-labeling here as the outer level. In particular, we enhance the accuracy of soft-labels $\hat{\boldsymbol{y}}^S$ by minimizing the mean squared error on the static dataset $\mathcal{D} = \{(\boldsymbol{x}_i, y_i)\}_{i=1}^N$,

$$\hat{\boldsymbol{y}}^{S'} = \hat{\boldsymbol{y}}^S - \frac{\lambda}{N} \frac{\partial \sum_{i=1}^N (f_{\boldsymbol{\theta}(\hat{\boldsymbol{y}})^S}^A(\boldsymbol{x}_i) - y_i)^2}{\partial (\hat{\boldsymbol{y}}^S)^\top} , \quad (9)$$

where $\lambda$ represents the soft-labeling learning rate. The nested optimization problem can be easily solved by *higher*, a library for higher-order optimization [19]. Once optimized, the labels are fed back to the first module, adaptively mentoring the proxy $f_{\boldsymbol{\theta}}^A(\cdot)$ according to Eq.(8). The same procedure can be employed for proxies $f_{\boldsymbol{\theta}}^B(\cdot)$ and $f_{\boldsymbol{\theta}}^C(\cdot)$, ultimately resulting in a more robust ensemble. We further clarify the novelty of this module in Appendix A.2.

## 4 Experiments

We conduct extensive experiments on design-bench to investigate the effectiveness and robustness of the proposed method. In Section 4.4, we benchmark our approach against several well-established baselines. In Section 4.5, we verify the effectiveness of two modules: *voting-based pairwise supervision* and *adaptive soft-labeling*, as well as other contributing factors. In Section 4.6, we investigate the sensitivity of our method to hyperparameter changes. While our primary focus is the *tri-mentoring* with 3 proxies, we additionally explore other *parallel-mentoring* implementations utilizing 5, 7, 9, and 11 proxies in Appendix A.1.

### 4.1 Task Overview

We adopt the design-bench which comprises both continuous and discrete tasks. Below, we outline the dataset details and evaluation protocols.

**Dataset.** We conduct experiments on four continuous tasks: **(1)** Superconductor (SuperC) [2]: discover an 86-D superconductor to maximize critical temperature with 17010 designs. **(2)** Ant Morphology (Ant) [20]: identify a 60-D ant morphology to crawl quickly with 10004 designs. **(3)** D'Kitty Morphology (D'Kitty) [21]: determine a 56-D D'Kitty morphology to crawl quickly with 10004 designs. **(4)** Hopper Controller (Hopper) [1]: find a neural network policy with 5126 weights to maximize return with 3200 designs. Besides, we perform experiments on four discrete tasks: **(1)** TF Bind 8 (TFB8) [5]: design a length 8 DNA sequence to maximize binding activity score with 32896 designs. **(2)** TF Bind 10 (TFB10) [5]: find a length 10 DNA sequence to maximize binding

activity score with 50000 designs. **(3)** NAS [1]: find a 64-D NN with 5 categories per dimension to maximize the performance on CIFAR10 with 1771 designs.

**Evaluation.** We use the oracle evaluation of design-bench to evaluate a certain design and the details of the oracle functions are reported in *Design-Bench Benchmark Tasks* in [1]. Following [7], we select the top N = 128 candidates for each method and report the $100^{th}$ percentile (maximum) normalized ground truth score. The score, denoted as $y_n$ is computed as $\frac{y-y_{min}}{y_{max}-y_{min}}$ where $y$ is the design score, and $y_{min}$ and $y_{max}$ are the lowest and highest scores in the complete unobserved dataset, respectively. In addition, we provide the $50^{th}$ percentile (median) normalized ground truth scores used in the prior work in Appendix A.3. We also provide the mean and median ranks of all comparison methods to better assess the overall performance.

## 4.2 Comparisons with Other Methods

In this paper, we benchmark our method against both gradient-based and non-gradient-based approaches. The gradient-based methods include: **(1)** Grad: optimizes the design against the learned proxy via simple gradient ascent; **(2)** DE (Deep Ensemble)[7]: optimizes the design against the mean ensemble of three proxies via gradient ascent; **(3)** GB (Gradient Boosting) [22]: sequentially trains new proxies to obtain a robust proxy, followed by gradient ascent using the proxy; **(4)** COMs [7]: lower bounds the proxy's prediction on out-of-distribution designs and subsequently carries out gradient ascent; **(5)** ROMA [23]: incorporates the smoothness prior into the proxy and optimizes the design against the proxy; **(6)** NEMO [24]: leverages normalized maximum likelihood to constrain the distance between the proxy and the ground-truth, and acquires new designs by gradient ascent; **(7)** BDI [25]: proposes to distill the information from the static dataset into the high-scoring design; **(8)** IOM [26]: enforces the invariance between the representations of the static dataset and generated designs to achieve a natural trade-off. Since our *tri-mentoring* adopts three proxies, methods using one proxy including COMs, ROMA and IOM are equipped with three proxies for a fair comparison. We also explore combining our method with ROMA and COMs and please refer to Appendix A.4 for an in-depth discussion and corresponding empirical results.

The non-gradient-based methods include: **(1)** BO-qEI [27]: fits a Gaussian Process, proposes candidate designs utilizing the quasi-Expected Improvement acquisition function, and assigns labels to the candidates with the proxy; **(2)** CMA-ES [28]: labels the sampled designs and gradually adapts the covariance matrix distribution towards the high-scoring part among the sampled designs; **(3)** REINFORCE [29]: parameterizes a design distribution and optimizes the distribution towards the optimal design by policy gradient; **(4)** CbAS [30]: trains a VAE model and progressively adapts the model to focus on the high-scoring designs; **(5)** Auto.CbAS [31]: retrains the proxy adopted in CbAS by leveraging importance sampling; **(6)** MIN [32]: learns an inverse map from a score to a design and then samples from the map conditioned an optimal score value.

## 4.3 Training Details

We follow the settings in [7, 25] if not specified. We adopt a three-layer MLP network with the ReLU function as the activation. We train the MLP model on the static dataset with a $1 \times 10^{-3}$ learning rate and an Adam optimizer. The fine-tuning learning rate $\gamma$ is set as $1 \times 10^{-3}$ and the soft-labeling learning rate $\lambda$ is set as $1 \times 10^{-1}$. The standard deviation $\delta$ is set as $1 \times 10^{-1}$ and the number of the samples $K$ is set as 10. All experiments are performed on a single V100 GPU. To ensure the robustness of our results, we perform 16 trials for each setting and report the mean and standard error. We've detailed the training time and computational overhead of our approach in Appendix A.5 to provide a comprehensive view of its practicality.

## 4.4 Results and Analysis

In Table 1 and Table 2, we present the results of our experiments for continuous and discrete tasks, respectively. A delineating line is drawn to separate the gradient-based methods from the non-gradient-based methods. Results for non-gradient-ascent methods are taken from [1]. The highest score of the static dataset for each task is denoted by $\mathcal{D}(\textbf{best})$. For each task, we highlight the algorithms that fall within one standard deviation of the highest performance by bolding their results.

**Continuous tasks.** (1) Table 1 demonstrates *tri-mentoring* attains the best results across the board, highlighting its effectiveness. Its consistent gains over Grad confirm its ability to tackle the out-of-distribution issue. We further test our tri-mentoring for out-of-distribution issues as detailed in Appendix A.6. (2) Compared to DE and GB, which also use multiple proxies, *tri-mentoring* achieves

Table 1: Results (maximum normalized score) on continuous tasks.

| Method | Superconductor | Ant Morphology | D'Kitty Morphology | Hopper Controller |
|---|---|---|---|---|
| $\mathcal{D}(\textbf{best})$ | 0.399 | 0.565 | 0.884 | 1.000 |
| BO-qEI | $0.402 \pm 0.034$ | $0.819 \pm 0.000$ | $0.896 \pm 0.000$ | $0.550 \pm 0.018$ |
| CMA-ES | $0.465 \pm 0.024$ | $\textbf{1.214} \pm \textbf{0.732}$ | $0.724 \pm 0.001$ | $0.604 \pm 0.215$ |
| REINFORCE | $0.481 \pm 0.013$ | $0.266 \pm 0.032$ | $0.562 \pm 0.196$ | $-0.020 \pm 0.067$ |
| CbAS | $\textbf{0.503} \pm \textbf{0.069}$ | $0.876 \pm 0.031$ | $0.892 \pm 0.008$ | $0.141 \pm 0.012$ |
| Auto.CbAS | $0.421 \pm 0.045$ | $0.882 \pm 0.045$ | $0.906 \pm 0.006$ | $0.137 \pm 0.005$ |
| MIN | $0.499 \pm 0.017$ | $0.445 \pm 0.080$ | $0.892 \pm 0.011$ | $0.424 \pm 0.166$ |
| Grad | $0.495 \pm 0.011$ | $0.934 \pm 0.011$ | $0.944 \pm 0.017$ | $1.797 \pm 0.116$ |
| DE | $\textbf{0.514} \pm \textbf{0.015}$ | $0.937 \pm 0.016$ | $\textbf{0.956} \pm \textbf{0.014}$ | $1.805 \pm 0.105$ |
| GB | $0.496 \pm 0.012$ | $0.926 \pm 0.029$ | $0.948 \pm 0.012$ | $\textbf{1.793} \pm \textbf{0.429}$ |
| COMs | $0.491 \pm 0.028$ | $0.856 \pm 0.040$ | $0.938 \pm 0.015$ | $0.642 \pm 0.167$ |
| ROMA | $0.508 \pm 0.014$ | $0.914 \pm 0.029$ | $0.930 \pm 0.012$ | $\textbf{1.728} \pm \textbf{0.266}$ |
| NEMO | $0.502 \pm 0.002$ | $\textbf{0.958} \pm \textbf{0.011}$ | $0.954 \pm 0.007$ | $0.481 \pm 0.003$ |
| IOM | $\textbf{0.522} \pm \textbf{0.018}$ | $0.926 \pm 0.030$ | $0.943 \pm 0.012$ | $1.015 \pm 0.380$ |
| BDI | $0.513 \pm 0.000$ | $0.906 \pm 0.000$ | $0.919 \pm 0.000$ | $\textbf{1.993} \pm \textbf{0.000}$ |
| *Tri-mentoring* | $\textbf{0.514} \pm \textbf{0.018}$ | $\textbf{0.948} \pm \textbf{0.014}$ | $\textbf{0.966} \pm \textbf{0.010}$ | $\textbf{1.983} \pm \textbf{0.110}$ |

Table 2: Results (maximum normalized score) on discrete tasks & ranking on all tasks.

| Method | TF Bind 8 | TF Bind 10 | NAS | Rank Mean | Rank Median |
|---|---|---|---|---|---|
| $\mathcal{D}(\textbf{best})$ | 0.439 | 0.467 | 0.436 | | |
| BO-qEI | $0.798 \pm 0.083$ | $0.652 \pm 0.038$ | $\textbf{1.079} \pm \textbf{0.059}$ | 10.1/15 | 11/15 |
| CMA-ES | $\textbf{0.953} \pm \textbf{0.022}$ | $0.670 \pm 0.023$ | $0.985 \pm 0.079$ | 6.4/15 | 4/15 |
| REINFORCE | $\textbf{0.948} \pm \textbf{0.028}$ | $0.663 \pm 0.034$ | $-1.895 \pm 0.000$ | 11.4/15 | 15/15 |
| CbAS | $\textbf{0.927} \pm \textbf{0.051}$ | $0.651 \pm 0.060$ | $0.683 \pm 0.079$ | 9.1/15 | 9/15 |
| Auto.CbAS | $0.910 \pm 0.044$ | $0.630 \pm 0.045$ | $0.506 \pm 0.074$ | 11.6/15 | 11/15 |
| MIN | $0.905 \pm 0.052$ | $0.616 \pm 0.021$ | $0.717 \pm 0.046$ | 11.0/15 | 12/15 |
| Grad | $0.886 \pm 0.035$ | $0.647 \pm 0.021$ | $0.624 \pm 0.102$ | 7.9/15 | 9/15 |
| DE | $0.900 \pm 0.056$ | $0.659 \pm 0.033$ | $0.655 \pm 0.059$ | 5.3/15 | 4/15 |
| GB | $\textbf{0.922} \pm \textbf{0.050}$ | $0.630 \pm 0.041$ | $0.716 \pm 0.088$ | 7.6/15 | 6/15 |
| COMs | $0.496 \pm 0.065$ | $0.622 \pm 0.003$ | $0.783 \pm 0.029$ | 10.0/15 | 11/15 |
| ROMA | $0.917 \pm 0.039$ | $0.672 \pm 0.035$ | $0.927 \pm 0.071$ | 5.7/15 | 6/15 |
| NEMO | $0.943 \pm 0.005$ | $\textbf{0.711} \pm \textbf{0.021}$ | $0.737 \pm 0.010$ | 5.0/15 | 4/15 |
| IOM | $0.861 \pm 0.079$ | $0.647 \pm 0.027$ | $0.559 \pm 0.081$ | 7.9/15 | 7/15 |
| BDI | $0.870 \pm 0.000$ | $0.605 \pm 0.000$ | $0.722 \pm 0.000$ | 8.1/15 | 9/15 |
| *Tri-mentoring* | $\textbf{0.970} \pm \textbf{0.001}$ | $\textbf{0.722} \pm \textbf{0.017}$ | $0.759 \pm 0.102$ | $\textbf{2.1/15}$ | $\textbf{2/15}$ |

better results in all four tasks, indicating its improved robustness by sharing ranking supervision signals among proxies. (3) DE typically outperforms simple gradient ascent due to ensemble prediction robustness, consistent with findings in [1]. (4) Other gradient-based methods, such as COMs, fail to achieve the performance as *tri-mentoring*, further highlighting its superior effectiveness. (5) In low-dimensional TF Bind 8 tasks, gradient-based methods (average rank 8.8) underperform compared to non-gradient-based methods (average rank 6.8); however, in high-dimensional TF Bind 10 tasks, gradient-based methods (average rank 7.7) surpass non-gradient-based methods (average rank 8.2). This suggests non-gradient-based methods, like REINFORCE and generative modeling, are more suited for low-dimensional design due to their global search ability, while gradient-based methods provide more direct guidance in high-dimensional designs.

**Discrete tasks.** (1) Table 2 shows that *tri-mentoring* achieves the best results in two out of the three tasks, with a marginal difference in the third. This indicates that *tri-mentoring* is a potent method for discrete tasks as well. (2) However, in complex tasks such as NAS, where each design is represented as a 64-length sequence of 5-category one-hot vectors, the performance of *tri-mentoring* is slightly compromised. This could be attributed to the encoding in design-bench not fully accounting for the sequential and hierarchical nature of network architectures, leading to less effective gradient updates. Our proposed method, also demonstrates its effectiveness on high-dimensional biological sequence design, achieving maximum normalized scores of 0.865 and 0.699 on GFP and UTR respectively.

**In summary,** *tri-mentoring* achieves the highest ranking as shown in Table 2 and Figure 3, and delivers the best performance in six out of the seven tasks.

Table 3: Ablation studies on *tri-mentoring*.

| Task | *tri-mentoring* | w/o *voting-based ps* | w/o *ada soft-labeling* | w/o *neighbor* | *post selection* |
|---|---|---|---|---|---|
| SuperC | $0.514 \pm 0.018$ | $0.505 \pm 0.014$ | $0.504 \pm 0.014$ | $\mathbf{0.516 \pm 0.017}$ | $0.512 \pm 0.011$ |
| Ant | $\mathbf{0.948 \pm 0.014}$ | $0.938 \pm 0.021$ | $0.945 \pm 0.018$ | $0.945 \pm 0.012$ | $0.945 \pm 0.009$ |
| D'Kitty | $0.966 \pm 0.010$ | $0.956 \pm 0.010$ | $0.947 \pm 0.008$ | $0.958 \pm 0.008$ | $\mathbf{0.970 \pm 0.013}$ |
| Hopper | $\mathbf{1.983 \pm 0.110}$ | $1.902 \pm 0.138$ | $1.916 \pm 0.108$ | $1.839 \pm 0.112$ | $1.901 \pm 0.148$ |
| TF Bind 8 | $0.970 \pm 0.001$ | $\mathbf{0.971 \pm 0.003}$ | $0.944 \pm 0.026$ | $0.950 \pm 0.018$ | $0.949 \pm 0.006$ |
| TF Bind 10 | $\mathbf{0.722 \pm 0.017}$ | $0.694 \pm 0.030$ | $0.710 \pm 0.025$ | $0.643 \pm 0.009$ | $0.635 \pm 0.027$ |
| NAS | $\mathbf{0.759 \pm 0.102}$ | $0.509 \pm 0.074$ | $0.538 \pm 0.082$ | $0.666 \pm 0.089$ | $0.519 \pm 0.076$ |

## 4.5 Ablation Studies

In this subsection, the proposed method serves as the baseline, and we systematically remove each module including *voting-based pairwise supervision* and *adaptive soft-labeling* to verify its contribution. The results are presented in Table 3. Besides the performance results here, we also provide an evaluation of the accuracy of generated pairwise labels to further verify the effectiveness of the two modules in Appendix A.7.

**Voting-based pairwise supervision.** Instead of using the proposed module, we compute the mean prediction of the ensemble and use this prediction to create pairwise consensus labels. We denote this as w/o *voting-based ps*. Our results in Table 3 show a decline in performance when adopting this alternative. A plausible explanation for this performance decline is that the alternative module fails to effectively exploit weak ranking supervision signals from individual proxies, resulting in reduced information exchange and collaboration among the proxies compared to the proposed module.

**Adaptive soft-labeling.** We remove this module and resort to using one-hot consensus labels. The performance across all tasks generally deteriorates compared to the full *tri-mentoring*. A possible explanation for this is that this module ensures that fine-tuned proxies are optimized to perform well on the static dataset by introducing soft-labels, reducing the risk of overfitting to consensus labels derived from individual proxy predictions. This demonstrates the significance of the *adaptive soft-labeling* module in mitigating the effects of label noise and enhancing the ensemble performance.

Furthermore, we assess the impact of neighborhood sampling in our method and a post selection strategy related to our method, with results outlined in Table 3.

**Neighborhood sampling.** Typically, we sample $K$ neighborhood points at the optimization point $x_t$ for pairwise labels. When neighborhood sampling is excluded (*w/o neighbor*), labels are directly generated near the static dataset. The performance generally deteriorates for *w/o neighbor* in Table 3, due to the lack of local ranking information around the optimization point.

**Post selection.** We investigate a variant, *post selection*, where the mean of two proxies is used to select the third proxy's candidates. From Table 3, we find that this variant generally yields worse results compared to the full tri-mentoring. This finding suggests that using the ranking supervision signals directly for design selection is less effective than allowing proxies to exchange ranking supervision signals within the ensemble to produce a more robust ensemble.

## 4.6 Hyperparameter Sensitivity

In this section, we study the sensitivity of our method to hyperparameters, specifically the number of neighborhood samples $K$ and the number of optimization steps $T$, on two tasks, the continuous Ant and the discrete TFB8. We also discuss the standard deviation hyperparameter $\delta$ in Appendix A.8.

**Number of neighborhood samples ($K$).** We evaluate the performance of our method for different values of $K$, i.e., the number of neighborhood samples around the optimization point. We test $K$ values of 5, 10, 15, 20, and 25, with $K = 10$ being the default value used in this paper. The results are normalized by dividing them by the result obtained for $K = 10$. As shown in Figure 4, the performance of the tri-mentoring method is quite robust to changes in $K$ for both tasks.

**Number of optimization steps ($T$).** We also investigate the impact of the number of optimization steps on the performance of our method. As indicated in Figure 5, the method is robust to changes in the number of optimization steps for both Ant and TFB8 tasks.

In summary, our sensitivity analysis demonstrates that the *tri-mentoring* method is robust to variations in key hyperparameters, ensuring stable performance across a range of settings.

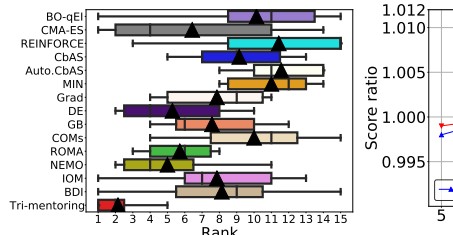 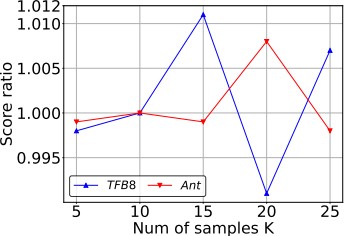 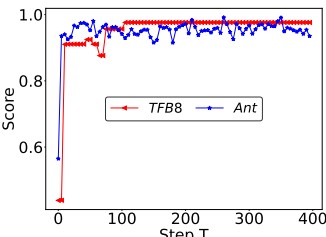

Figure 3: Whiskers signify rank min/max; medians and means are shown as vertical lines and black triangles, respectively.

Figure 4: **The ratio of** the performance of our *tri-mentoring* method with $K$ **to** the performance with $K = 10$.

Figure 5: The performance score as a function of the optimization step $T$ for both TFB8 and Ant tasks.

## 5  Related Work

**Offline model-based optimization.** Recently two broad groups of methods have emerged for offline model-based optimization. One group is based on generative modeling, where methods aim to gradually adapt a generative model towards the high-scoring design [31–33]. Another group is based on using gradient ascent to optimize existing designs via gradient information. These methods generally try to incorporate prior information into the proxy before using it for gradient ascent. Examples of this approach include COMs [7], ROMA [23], NEMO [24], BDI [25, 34] and IOM [26]. Our proposed method called *parallel-mentoring* (*tri-mentoring*) belongs to this category. We maintain an ensemble of proxies and aim to incorporate weak ranking signals from any pair of proxies into the third. This symmetric learning process, which cycles through all proxies, enhances the robustness and resilience of our ensemble. Our proposed ensemble training process, with its focus on parallel-mentoring, has the potential to improve the proxy and reward training [35, 33], thereby contributing to advancements in biological sequence design. Notably, the contemporaneous ICT method [36] exchanges direct proxy scores, which may be less robust than our pairwise comparison approach.

**Tri-training.** Our work is related to tri-training [37] which trains three classifiers and refines them using unlabeled samples. In each round, an unlabeled sample is labeled for one classifier if the other two agree on the label, under specific conditions. While *tri-mentoring* is inspired by tri-training, there are fundamental differences between them: (1) Tri-training aims to enhance classification tasks by mitigating label noise, while *tri-mentoring* focuses on producing a more robust regression ensemble; (2) Tri-training leverages unlabeled data, while *tri-mentoring* operates on samples near the current optimization point; (3) *Tri-mentoring* incorporates an *adaptive soft-labeling* module to mitigate label noise, which is not present in tri-training. In addition to tri-training and *tri-mentoring*, there are other research works [38] [39] [40] involving multiple proxies cooperating to improve learning. Unlike tri-training and *tri-mentoring* with three proxies, these methods focus on two proxies.

**Bi-level optimization for hyperparameter optimization.** Bi-level optimization has become increasingly popular for hyperparameter optimization [41–46] due to the hierarchical problem structure [47]. In the inner level, the relationship between model parameters and hyperparameters is established by minimizing the training loss. Meanwhile, the outer level optimizes hyperparameters through the connection built at the inner level by minimizing the validation loss. A specific category of hyperparameters is soft-label [48, 49], which is updated under the guidance of noise-free data to reduce noise. In this paper, we propose *adaptive soft-labeling* to reduce noise in the consensus labels.

**Ensemble learning.** Ensemble learning techniques train multiple base learners and aggregate their predictions to achieve better performance and generalization than individual base learners alone [8, 9, 50, 10, 11, 51, 52]. These methods can be broadly classified into boosting [53, 54], bagging [55], and stacking [56]. In contrast to our proposed *tri-mentoring*, where multiple proxies collaborate to enhance the learning process, ensemble learning mainly involves interaction during the aggregation phase, without influencing each other's training.

**Pairwise learning to rank.** Learning to rank [16, 57] has been extensively employed by commercial search engines for ranking search results [58, 59]. Unlike pointwise methods that score inputs, pairwise methods focus on relative order, aligning more with ranking concepts [57]. Recent research [60] applies pairwise binary cross-entropy loss for training reward models in reinforcement learning, a technique used in ChatGPT [18]. Our work expresses a proxy's ranking ability through pairwise comparison labels, generating consensus labels via majority voting to enable mutual mentoring.

# 6 Conclusion

In this work, we introduce *parallel-mentoring* to enhance ensemble robustness against out-of-distribution issues through mutual mentoring among proxies. Focusing on a three-proxy case, we instantiate this as *tri-mentoring*, with two modules: *voting-based pairwise supervision* for generating consensus labels, and *adaptive soft-labeling* which mitigates label noise through bi-level optimization. Experimental results validate our approach's effectivenss. We discuss potential negative impacts in Appendix A.9 and limitations in Appendix A.10.

# 7 Acknowledgement

We thank CIFAR for support under the AI Chairs program. This research was empowered in part by the computational support provided by Compute Canada (www.computecanada.ca).

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

# A  Appendix

## A.1  Scenarios for Parallel Mentoring with Multiple Proxies

### A.1.1  Method

In the primary paper, we mainly focus on a scenario with three proxies. Here, we extend our method to incorporate $M$ proxies. We revisit the two essential modules.

**Voting-based pairwise supervision.** We train $M$ ($M \geq 3$) parallel proxies, $f_{\boldsymbol{\theta}}^1(\cdot)$, $f_{\boldsymbol{\theta}}^2(\cdot)$, $\cdots$, $f_{\boldsymbol{\theta}}^M(\cdot)$, each initialized differently, on the static dataset. Their mean is utilized as the final prediction:

$$f_{\boldsymbol{\theta}}(\cdot) = \frac{1}{M}(f_{\boldsymbol{\theta}}^1(\cdot) + f_{\boldsymbol{\theta}}^2(\cdot) + \cdots f_{\boldsymbol{\theta}}^M(\cdot)). \tag{10}$$

We generate the pairwise comparison labels $\hat{\boldsymbol{y}}^1$, $\hat{\boldsymbol{y}}^2$, $\cdots$, and $\hat{\boldsymbol{y}}^M$ for each proxy in the same way. We extend the subsequent majority voting part and derive the pairwise consensus labels $\hat{\boldsymbol{y}}^V$ via an element-wise majority voting:

$$\hat{\boldsymbol{y}}_{ij}^V = \text{majority\_voting}(\hat{\boldsymbol{y}}_{ij}^1, \hat{\boldsymbol{y}}_{ij}^2, \cdots, \hat{\boldsymbol{y}}_{ij}^M). \tag{11}$$

Here, $i$ and $j$ are the indexes of the neighborhood samples.

**Adaptive soft-labeling.** This module remains the same as it is designed for an individual proxy. We carry out fine-tuning and soft-labeling via bi-level optimization to adaptively mentor the proxy.

**Setting on $M$.** In Eq.(11), $M$ can be any positive number greater than 2 as a decision may not be reached with just two proxies. In this study, we consider $M$ as an odd number to ensure a decisive outcome in the voting process. Cases with an even number of proxies can be handled by adopting strategies like maintaining the original labels and skipping the fine-tuning step when the proxies are evenly split in their labels. However, we do not delve into these cases for brevity. We examine scenarios with $M$ equal to 3, 5, 7, 9, and 11.

### A.1.2  Experiments

We conduct experiments on the Ant task and the TFB8 task. The performance ratio comparing the performance of *parallel mentoring* to that of *tri-mentoring*, is computed as a function of $M$ (the number of proxies). The results are displayed in Figure 6.

(1) Our observations indicate that as the number of proxies ($M$) increases, the performance ratios for both tasks initially improve, eventually reaching a plateau. This behavior suggests that an increased number of proxies enhances the robustness of the ensemble due to the increased diversity. However, this impact lessens as the number of proxies increases further, with the ensemble's robustness plateauing after a certain point. (2) Somewhat unexpectedly, the performance with $M = 7$ shows a slight dip on the Ant task. A possible explanation for this could be the dynamics of the voting system. When we have $M = 3$, some voting happens when two proxies agree but conflict with the third. However, when $M$ increases to 7, voting may occur when four proxies align with one another but dissent with the remaining three. Such a scenario can make consensus labels less reliable, potentially explaining the poor performance of the $M = 7$ case on the Ant task. (3) Finally, it's important to note that adding more proxies also amplifies computational complexity. This increase could become a restricting factor when trying to scale the method to include a larger number of proxies.

## A.2  Novelty of Adaptive Soft-labeling

Our adaptive soft-labeling approach, while bearing some superficial resemblance to existing bi-level optimization methods applied to soft-labels, possesses unique characteristics that underscore its innovation. In this appendix, we detail these distinctive facets.

**Initialization:** Contrary to previous works that aim to address label noise in a collected dataset, our method handles label noise created by a unique voting mechanism along the optimization path.

**Optimization:** Previous work typically leverages the classification loss of a clean validation set to update the soft-label. However, our method proposes an innovative strategy of employing the regression loss of the static dataset for soft-label updates. This unique strategy is due to our novel

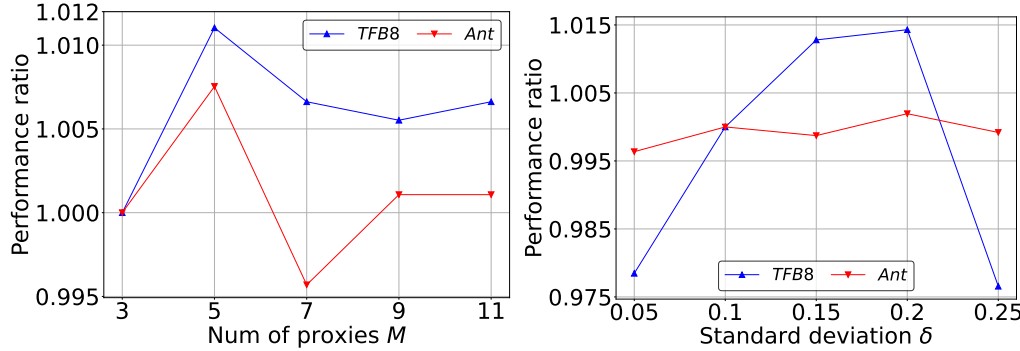

Figure 6: **Ratio of** performance with $M$ proxies **to** performance with $M = 3$ proxies.

Figure 7: **Ratio of** performance with standard deviation $\delta$ **to** performance with $\delta = 0.10$.

Table 4: Results (median normalized score) on continuous tasks.

| Method | Superconductor | Ant Morphology | D'Kitty Morphology | Hopper Controller |
|---|---|---|---|---|
| $\mathcal{D}(\textbf{best})$ | 0.399 | 0.565 | 0.884 | 1.000 |
| BO-qEI | $0.300 \pm 0.015$ | $0.567 \pm 0.000$ | $0.883 \pm 0.000$ | $0.343 \pm 0.010$ |
| CMA-ES | $0.379 \pm 0.003$ | $-0.045 \pm 0.004$ | $0.684 \pm 0.016$ | $-0.033 \pm 0.005$ |
| REINFORCE | $\textbf{0.463} \pm \textbf{0.016}$ | $0.138 \pm 0.032$ | $0.356 \pm 0.131$ | $-0.064 \pm 0.003$ |
| CbAS | $0.111 \pm 0.017$ | $0.384 \pm 0.016$ | $0.753 \pm 0.008$ | $0.015 \pm 0.002$ |
| Auto.CbAS | $0.131 \pm 0.010$ | $0.364 \pm 0.014$ | $0.736 \pm 0.025$ | $0.019 \pm 0.008$ |
| MIN | $0.336 \pm 0.016$ | $\textbf{0.618} \pm \textbf{0.040}$ | $\textbf{0.887} \pm \textbf{0.004}$ | $0.352 \pm 0.058$ |
| Grad | $0.339 \pm 0.015$ | $0.564 \pm 0.014$ | $0.877 \pm 0.005$ | $0.384 \pm 0.004$ |
| DE | $0.333 \pm 0.004$ | $0.570 \pm 0.011$ | $0.875 \pm 0.004$ | $0.385 \pm 0.007$ |
| GB | $0.373 \pm 0.013$ | $0.550 \pm 0.021$ | $0.869 \pm 0.009$ | $0.374 \pm 0.008$ |
| COMs | $0.316 \pm 0.022$ | $0.568 \pm 0.002$ | $0.883 \pm 0.002$ | $0.346 \pm 0.009$ |
| ROMA | $0.368 \pm 0.019$ | $0.475 \pm 0.036$ | $0.856 \pm 0.008$ | $0.388 \pm 0.007$ |
| NEMO | $0.322 \pm 0.008$ | $0.593 \pm 0.000$ | $0.885 \pm 0.000$ | $0.361 \pm 0.001$ |
| IOM | $0.348 \pm 0.022$ | $0.516 \pm 0.037$ | $0.876 \pm 0.007$ | $0.368 \pm 0.008$ |
| BDI | $0.412 \pm 0.000$ | $0.474 \pm 0.000$ | $0.855 \pm 0.000$ | $\textbf{0.408} \pm \textbf{0.000}$ |
| *tri-mentoring* | $0.355 \pm 0.003$ | $0.606 \pm 0.007$ | $\textbf{0.886} \pm \textbf{0.001}$ | $0.391 \pm 0.004$ |

recognition that, despite their differing data distributions, the soft-labels and the static dataset share significant underlying similarities. They both represent the same ground-truth from pairwise and pointwise perspectives, respectively.

**Objective:** Whereas previous works primarily focus on mitigating label noise to enhance the performance of a classification model, our work takes a distinctive route by addressing label noise to improve the performance of a regression model for offline MBO. This adaptation to a new context further underscores the innovative aspects of our approach.

### A.3 Additional Results on 50th Percentile Scores

In the main paper, we presented the $100^{th}$ percentile scores. Here, we offer supplementary results on the $50^{th}$ percentile scores, which have been previously utilized in the design-bench work [1], to further validate the efficacy of *tri-mentoring*. Continuous task results can be found in Table 4 while discrete task results and ranking statistics are shown in Table 5. A review of Table 5 reveals that *tri-mentoring* achieves the highest ranking, demonstrating its effectiveness in this context.

### A.4 Integration of COMs and ROMA with Tri-mentoring

We explore the integration of ROMA and COMs with our tri-mentoring approach. The results in Table 6 offer insights into the outcomes when combining these methods on the Ant and TFB8 tasks.

When COMs are combined with tri-mentoring, a slight drop in performance is observed. This decrease might originate from COMs' propensity to score lower for out-of-distribution designs, potentially conflicting with tri-mentoring's majority voting mechanism for such designs.

Table 5: Results (median normalized score) on discrete tasks & ranking on all tasks.

| Method | TF Bind 8 | TF Bind 10 | NAS | Rank Mean | Rank Median |
|---|---|---|---|---|---|
| $\mathcal{D}$(**best**) | 0.439 | 0.467 | 0.436 | | |
| BO-qEI | $0.439 \pm 0.000$ | $0.467 \pm 0.000$ | $0.544 \pm 0.099$ | 8.0/15 | 8/15 |
| CMA-ES | $0.537 \pm 0.014$ | $0.484 \pm 0.014$ | $0.591 \pm 0.102$ | 8.0/15 | 5/15 |
| REINFORCE | $0.462 \pm 0.021$ | $0.475 \pm 0.008$ | $-1.895 \pm 0.000$ | 10.6/15 | 14/15 |
| CbAS | $0.428 \pm 0.010$ | $0.463 \pm 0.007$ | $0.292 \pm 0.027$ | 12.7/15 | 12/15 |
| Auto.CbAS | $0.419 \pm 0.007$ | $0.461 \pm 0.007$ | $0.217 \pm 0.005$ | 13.3/15 | 13/15 |
| MIN | $0.421 \pm 0.015$ | $0.468 \pm 0.006$ | $0.433 \pm 0.000$ | 7.7/15 | 9/15 |
| Grad | $0.532 \pm 0.017$ | $\mathbf{0.529 \pm 0.027}$ | $0.443 \pm 0.126$ | 6.1/15 | 6/15 |
| DE | $\mathbf{0.581 \pm 0.034}$ | $\mathbf{0.534 \pm 0.014}$ | $0.474 \pm 0.085$ | 5.4/15 | 4/15 |
| GB | $0.503 \pm 0.054$ | $0.455 \pm 0.020$ | $\mathbf{0.559 \pm 0.090}$ | 7.3/15 | 6/15 |
| COMs | $0.439 \pm 0.000$ | $0.466 \pm 0.002$ | $0.529 \pm 0.003$ | 7.9/15 | 8/15 |
| ROMA | $0.548 \pm 0.017$ | $\mathbf{0.516 \pm 0.020}$ | $0.529 \pm 0.008$ | 5.7/15 | 5/15 |
| NEMO | $0.439 \pm 0.018$ | $0.456 \pm 0.015$ | $\mathbf{0.568 \pm 0.021}$ | 7.0/15 | 8/15 |
| IOM | $0.437 \pm 0.010$ | $0.475 \pm 0.010$ | $-0.083 \pm 0.012$ | 9.0/15 | 7/15 |
| BDI | $0.439 \pm 0.000$ | $0.476 \pm 0.000$ | $0.517 \pm 0.000$ | 6.6/15 | 7/15 |
| *tri-mentoring* | $\mathbf{0.609 \pm 0.021}$ | $\mathbf{0.527 \pm 0.008}$ | $0.516 \pm 0.028$ | **3.4/15** | **2/15** |

Table 6: Integration results for tri-mentoring with COMs and ROMA.

| Method | Ant | TFB8 |
|---|---|---|
| COMs | 0.856 | 0.496 |
| Tri-mentoring | 0.948 | 0.970 |
| COMs + Tri-mentoring | 0.941 | 0.960 |
| ROMA | 0.914 | 0.917 |
| ROMA + Tri-mentoring | 0.945 | 0.971 |

Conversely, ROMA inherently promotes smoother proxy models, a characteristic theoretically orthogonal to our tri-mentoring method. The presented results hint that our current benchmarks might not be adequately challenging to fully exploit the combined strengths of ROMA and tri-mentoring.

## A.5 Training Time Analysis

Understanding the time complexity and computational overhead of a method is crucial for practical implementations. Hence, we provide detailed estimates for the training time of our approach.

For a single proxy training on the static dataset, the training time required is 101.1 s for TFB8 and 33.5 s for Ant. This training duration remains consistent across all methods employing a proxy. The added time overhead in tri-mentoring arises mainly from the proxy fine-tuning step. Specifically, the fine-tuning step takes 0.21 s for TFB8 and 0.08 s for Ant.

Given the minimal overhead of our method, especially when considering its benefits, we believe it presents a feasible and efficient choice for researchers and practitioners aiming for effective results without compromising on speed.

## A.6 Validation of Tri-mentoring against Out-of-distribution Issues

To substantiate our tri-mentoring approach's competence in addressing out-of-distribution (OOD) challenges, we have performed additional experiments on TFBind8 and TFBind10. We choose the two tasks as we could easily identify high-scoring designs and thereby create an out-of-distribution test set. From each dataset, we sample 1000 high-scoring designs to form our OOD test sets. For these, our tri-mentoring ensemble have been tested over 5 runs against a mean ensemble and a single proxy, and the mean squared error (MSE) is used as the metric. These results in Table 7 illustrate that the tri-mentoring ensemble consistently outperforms the other models in handling OOD designs.

## A.7 Accuracy of Pairwise Consensus Labels

In addition to the performance results presented in the main paper, we also examine the accuracy of the optimized consensus labels $\hat{y}^{S'}$. This analysis further substantiates the effectiveness of our

Table 7: MSE Results for OOD Validation using TFBind8 and TFBind10 Datasets

| Model | TFBind8 (MSE) | TFBind10 (MSE) |
|---|---|---|
| Tri-mentoring ensemble | $0.0537 \pm 0.0008$ | $0.2271 \pm 0.0075$ |
| Mean ensemble | $0.0549 \pm 0.0006$ | $0.2669 \pm 0.0084$ |
| Single proxy | $0.0660 \pm 0.0012$ | $0.2776 \pm 0.0185$ |

method. For the D'Kitty and TFB8 tasks, we utilize the ground-truth function to determine the ground-truth pairwise labels. This enables us to assess the accuracy of $\hat{y}^{S'}$. For easier accuracy computation, these labels are converted into one-hot labels.

(1) Recall that the pairwise comparison labels of a single proxy serve as its ranking supervision signals. In our analysis, we found that for a single proxy, 13.45% of pairwise comparison labels for the D'Kitty task and 8.38% for the TFB8 task differ from the optimized consensus labels $\hat{y}^{S'}$. This reveals the extent to which our method modifies the original labels. (2) Further analysis shows that, of the conflicting optimized labels, 62.91% are accurate for D'Kitty and 63.16% are accurate for TFB8. These results reinforce the overall efficacy of our method. (3) When we remove the *voting-based pairwise supervision* module, we note a decrease in accuracy from 62.91% to 52.21% for D'Kitty and from 63.16% to 55.63% for TFB8. Similarly, omitting the *adaptive soft-labeling* module leads to a drop in accuracy from 62.91% to 57.16% for D'Kitty and from 63.16% to 60.86% for TFB8. These experiments underscore the crucial role of both modules in preserving the label accuracy.

## A.8 Additional Analysis on Sensitivity to the Standard Deviation Hyperparameter

We further delve into how the standard deviation hyperparameter $\delta$ in neighborhood sampling, impacts the performance of our method. We experiment with $\delta$ values of 0.05, 0.10, 0.15, 0.20, and 0.25, with 0.10 being the default value employed in this paper. The results are normalized by dividing them by the result obtained for $\delta = 0.10$. As demonstrated in Figure 7, *tri-mentoring* exhibits remarkable robustness to variations in $\delta$ for both the continuous Ant and the discrete TFB8 tasks.

We further explore $\gamma$ and $\lambda$. We conduct additional experiments over varied ranges on TFB8 and Ant, evaluating across the following values:

- $\gamma$: $[2.5e - 4, 5e - 4, 1e - 3, 2e - 3, 4e - 3]$
- $\lambda$: $[0.025, 0.05, 0.1, 0.2, 0.4]$

These results in Table 8 confirm the method's resilience against hyperparameter variations on $\gamma$ and $\lambda$, emphasizing its potential for real-world scenarios despite its perceived complexity.

Table 8: Sensitivity Analysis Results on TFB8 and Ant.

| Parameter | TFB8 | | | | | Ant | | | | |
|---|---|---|---|---|---|---|---|---|---|---|
| | 2.5e-4 | 5e-4 | 1e-3 | 2e-3 | 4e-3 | 2.5e-4 | 5e-4 | 1e-3 | 2e-3 | 4e-3 |
| $\gamma$ | 0.903 | 0.995 | 1.000 | 0.996 | 1.006 | 0.953 | 1.008 | 1.000 | 1.021 | 1.022 |
| $\lambda$ | 0.945 | 0.983 | 1.000 | 1.000 | 0.995 | 1.011 | 0.989 | 1.000 | 0.992 | 0.997 |

Last but not least, we explore the number of the fine-tuning step. We select a single step primarily for efficiency and our comprehensive experiments indicate that increasing the number of fine-tuning steps does not significantly impact the model's performance, confirming the method's robustness in this regard. For the TFB8 dataset, the normalized results for 1, 2, 3, and 4 fine-tuning steps are 1.000, 0.986, 1.006, and 0.992 respectively. Similarly, for the Ant dataset, they are 1.000, 0.994, 1.000, and 0.985. As seen, the performance remains consistent across different numbers of steps.

## A.9 Broader Impacts

Our work could potentially expedite the development of new materials, biomedical innovations, or robotics technologies, leading to significant advancements in these areas. However, as with all powerful tools, there are potential risks if misused. One potential negative impact could be the misuse of this technology in designing objects or entities for harmful purposes. For instance, in the wrong hands, the ability to optimize designs could be used to create more efficient weapons or harmful

biological agents. Therefore, it is crucial to implement appropriate safeguards and regulations on the use of such technology, particularly in sensitive areas.

### A.10 Limitations

Despite the promising results demonstrated by our method, its performance is largely dependent on the accuracy of the design encoding. For tasks of high complexity, such as Neural Architecture Search (NAS) - which represents each design as a 64-length sequence of 5-category one-hot vectors - the performance of *tri-mentoring* is somewhat limited. This shortfall could be due to the default encoding technique of design-bench [1], which may fail to adequately capture the sequential and hierarchical nature of neural architectures, leading to ineffective gradient updates. This challenge suggests that, while our method provides a general framework for offline model-based optimization, task-specific techniques might be necessary for effective design encoding, especially in the context of complex tasks. Potential future research could explore ways of integrating problem-specific knowledge into the design encoding process to address these complexities more effectively.

