# OpenReview forum: "Parallel-mentoring for Offline Model-based Optimization"
_NeurIPS.cc/2023/Conference — NeurIPS 2023 poster_

### Official Review · Reviewer_NEb9 · 2023-07-03

**Soundness:** 3 good
**Presentation:** 3 good
**Contribution:** 3 good
**Rating:** 7
**Confidence:** 5

**Summary:**

This paper introduces a novel offline model-based optimization (MBO) framework that incorporates a fine-tuning strategy for ensemble proxy models. Instead of directly using three independent proxy models, the paper suggests a "voting" strategy. In this scheme, based on the majority order of two (near) inputs, each model is fine-tuned using binary cross-entropy (BCE) loss before it is utilized for the optimization problem. The paper contends that this approach enables the proxy models to mentor each other, thereby leading to improved performance during the input optimization stage.

**Strengths:**

- The paper is well-written and easy to follow.
- The experimental results are evaluated extensively, as well as showing quite a good performance compared with prior offline MBO methods.
- To best of my knowledge, exploring fine-tuning strategy to improve the performance of ensemble models is relatively under-explored in offline MBO literature. In this respect, I think this paper investigates important but overlooked  problem.

**Weaknesses:**

- My primary concern revolves around hyperparameter selection. It appears the method introduces a considerable number of additional hyperparameters, including, but not limited to, learning rates for fine-tuning soft-labels and proxy models, the variance $\delta^2$ for selecting an adjacent input from the current solution, and the number of near inputs, $K$. Although the authors demonstrated the stability and effect of these hyperparameters (e.g., $K$), an analysis of some other hyperparameters, such as $\delta^2$ or fine-tuning learning rates, is still missing. Given that the paper addresses "offline" MBO, the experiments should detail how these hyperparameters can be determined (e.g., $\delta^2$ might be defined as a function of input dimension or a constant). Alternatively, the authors should demonstrate that the proposed method is robust to variations in these hyperparameters.
- The proposed fine-tuning strategy lacks the original regression objective; the model might produce inaccurate values at training points after fine-tuning.
- Although it's not a major weakness, any theoretical guarantee or supporting intuition would significantly enhance the manuscript.

**Questions:**

- The paper deals with discrete tasks as well, while the optimization strategy is based on gradient ascent. How the input optimization is performed?
- I wonder if the authors tried to incorporate the method on prior offline MBO method, e.g., COMs to pre-trained the model first and then fine-tune models via the proposed fine-tuning strategy.
- I wonder if the model is fine-tuned sequentially or not. Specifically, once  $f_A$ is fine-tuned, then is fine-tuned $f_A$ used for fine-tuning $f_B$ or the original $f_A$ before fine-tuning used?
- Why the fine-tuning step is set as 1? Did the authors try more steps?

**Limitations:**

The authors discussed the limitations and potential negative impact in their manuscript.

---

> ### Author Rebuttal · Authors · 2023-08-06
>
> ## Weaknesses
>
> > My primary concern revolves around hyperparameter selection.
>
> Please refer to the global response "On Additional Hyperparameters".
>
> > The proposed fine-tuning strategy lacks the original regression objective; the model might produce inaccurate values at training points after fine-tuning.
>
> Thank you for pointing out the potential pitfalls of our fine-tuning strategy concerning the original regression objective. Emphasizing soft-labels can indeed shift predictions at the original training points. However, our primary goal with adaptive soft-labeling is broader: it aims to enhance the model's generalization, especially towards out-of-distribution points around the current optimization point. This might sometimes sacrifice pinpoint accuracy on training points but significantly improves the model's ranking ability and adaptability beyond the training set.
>
> To emphasize, **our model still respects the regression objective**. The soft-labels $\boldsymbol{\hat{y}}^{S}$, integral to the fine-tuning, are tied to the static dataset and refined to minimize the mean squared error, as depicted in Eq.(9) of our paper:
>
> $$\boldsymbol{\hat{y}}^{S'} = \boldsymbol{\hat{y}}^{S} - \frac{\lambda}{N} \frac{\partial \sum_{{i=1}}^{N}(f^A_{\boldsymbol{\theta}(\boldsymbol{\hat{y}})^S}(\boldsymbol{x}_i) - y_i)^2}{\partial (\boldsymbol{\hat{y}}^{S}){^\top}}.$$
>
>
> In short, our model finds a balance between adaptability from soft-labeling and the foundational accuracy of the original regression, resulting in a versatile and robust ensemble for real-world scenarios.
>
> > Although it's not a major weakness, any theoretical guarantee or supporting intuition would significantly enhance the manuscript.
>
> We have illustrated the motivation of our paper in Figure 1. The underpinning idea behind tri-mentoring is to harness the voted ranking signal to fine-tune the proxy. This is clearly depicted in Figure 1 of our paper. Here, we maintain three parallel proxies, namely fA, fB, and fC. Suppose we have two designs x1 and x2 in the neighborhood of the current optimization point. If proxies fA and fB concur that the score of x1 is larger than that of x2, while fC disagrees, we follow the majority voting principle. In this scenario, the agreement between fA and fB provides a more trustworthy ranking. Their voted ranking signal fV(x1) > fV(x2) is then used to guide or 'mentor' fC, thereby enhancing its predictive performance.
>
> ## Questions
>
> > The paper deals with discrete tasks as well, while the optimization strategy is based on gradient ascent. How the input optimization is performed?
>
> You are correct to point out the discrete nature of the tasks we've addressed. For input optimization, we adopt an encoding-decoding strategy as per the design-bench [1]. This allows us to transform the discrete space into a continuous one that we can perform gradient ascent on. After obtaining optimized continuous values, they are decoded back to the original discrete space. This approach lets us effectively leverage the gradient ascent for optimization.
>
> > I wonder if the authors tried to incorporate the method on prior offline MBO method, e.g., COMs to pre-trained the model first and then fine-tune models via the proposed fine-tuning strategy.
>
> Thank you for the insightful inquiry on integrating conservatism (COMs) and Gaussian priors (ROMA) with our tri-mentoring.
>
> In brief, our additional experimental results on Ant and TFB8 tasks are:
>
> | Method | Ant | TFB8 |
> |--------|------|-----|
> | COMs | 0.856  | 0.496 |
> | Tri-mentoring | 0.948  | 0.970 |
> | COMs + Tri-mentoring | 0.941 | 0.960  |
> | ROMA | 0.914  | 0.917 |
> | Tri-mentoring | 0.948  | 0.970 |
> | ROMA + Tri-mentoring | 0.945  | 0.971 |
>
> While COMs combined with tri-mentoring leads to a slight drop in performance, the ROMA and tri-mentoring combination retains similar outcomes. The dip with COMs might arise from its lower scoring for out-of-distribution designs, potentially conflicting with the tri-mentoring's majority voting mechanism for odd designs.
>
> On the other hand, ROMA, promotes smooth proxy models. Theoretically, this characteristic should be orthogonal to our tri-mentoring. The absence of improvement might suggest that our current benchmark is not sufficiently challenging to truly exploit the combined strengths of both ROMA and tri-mentoring. We'll ensure to incorporate these analyses into Appendix.
>
> > I wonder if the model is fine-tuned sequentially or not. Specifically, once fA is fine-tuned, then is fine-tuned fA used for fine-tuning fB or the original fA before fine-tuning used?
>
> As for the fine-tuning process, it is not performed sequentially. Instead, we generate consensus labels using the three proxies, following which each proxy is independently fine-tuned using these consensus labels and the adaptive soft-labeling method.
>
> > Why the fine-tuning step is set as 1? Did the authors try more steps?
>
> Thank you for raising the question regarding the number of fine-tuning steps, specifically why we have set it to 1.
>
> We select a single step primarily for efficiency. Moreover, our comprehensive experiments indicate that increasing the number of fine-tuning steps does not significantly impact the model's performance, confirming the method's robustness in this regard. Below are our normalized results on TFB8 and Ant for 1, 2, 3, and 4 fine-tuning steps, respectively. These results are normalized by the performance with just one
>  step:
>
> - TFB8: [1.000, 0.986, 1.006, 0.992]
> - Ant: [1.000, 0.994, 1.000, 0.985]
>
> As seen, the performance remains consistent across different numbers of steps.
>
> ## Overall
>
> Have we addressed your concerns adequately? Your feedback is invaluable, and we await further discussion. Thank you.

---

> > ### Comment · Reviewer_NEb9 · 2023-08-11
> > **Response**
> >
> > Thank you for the detailed response. The provided response addressed most of my concerns well. Hence, I raise my score to 7. Please add the related discussion in the final manuscript if the paper is accepted.

---

> > > ### Author Response · Authors · 2023-08-11
> > > **Thanks for your prompt feedback and continued support.**
> > >
> > > Thank you for your prompt feedback and the adjusted score. We are glad the response addressed most of your concerns. Rest assured, the related discussion will be diligently added to the final manuscript, as suggested, if the paper is accepted.

---

### Official Review · Reviewer_VCj9 · 2023-07-05

**Soundness:** 2 fair
**Presentation:** 3 good
**Contribution:** 2 fair
**Rating:** 5
**Confidence:** 2

**Summary:**

The aim of the paper is to tackle the out-of-distribution problem in offline model-based optimization methods. To this end, the authors drew inspiration from recent studies that proposed better ensemble proxies and weak ranking supervision signals. They proposed a new approach called parallel-mentoring, which consists of two main modules. The first module, a voting-based pairwise supervision module, generates consensus labels. The second module, an adaptive soft-labeling module, aims to learn more accurate soft labels. The authors conducted experiments on both continuous and discrete tasks, and their approach achieved the highest ranking in the results.

**Strengths:**

1. The paper proposed a new approach, which settles the out-of-distribution problem in offline model-based optimization.

2. The paper is clear and easy to understand.

3. The paper conducts extensive experiments on the design-bench.


**Weaknesses:**

1.	The research question is poorly defined, and the statement of the importance of the research is lacking. The out-of-distribution problem is not clearly defined, and its significance is not convincingly conveyed.

2.	There is no experimental evidence to support the assertion that the tri-mentoring approach can effectively address the out-of-distribution problem.


**Questions:**

1.	How does leveraging the knowledge of the static dataset help to learn more accurate soft-labels?

**Limitations:**

The authors have adequately addressed the limitations.

---

> ### Author Rebuttal · Authors · 2023-08-06
>
> ## General Reply
>
> We greatly appreciate your detailed feedback and insights. We're committed to addressing each point raised thoroughly. Thank you.
>
> ## Weaknesses
>
> > The research question is poorly defined, and the statement of the importance of the research is lacking. The out-of-distribution problem is not clearly defined, and its significance is not convincingly conveyed.
>
> Our sincere thanks for your insightful comments. The primary research question of our work revolves around offline model-based optimization (MBO), a topic we detail in our "Preliminaries: Gradient Ascent on Offline Model-based Optimization" section. It aims to maximize a black-box objective function using a static dataset of designs and scores.
>
> We acknowledge your point regarding the necessity to highlight the importance of our research. As a remedy, we plan to emphasize this by adding the following text after the sentence "Designing new objects or entities to optimize specific properties is a widespread challenge, encompassing various domains such as materials, robots, and DNA sequences." in Line 25: "A more effective solution to this challenge holds enormous potential benefits for society, such as facilitating the proposal of new superconductors to reduce energy costs, enabling the design of faster robots for service, and advancing new drug discoveries to combat diseases."
>
> Regarding the out-of-distribution problem, it was our aim to illustrate its significance from Line 101 to Line 103: "However, this method faces a challenge with the proxy being vulnerable to out-of-distribution designs. When handling designs that substantially differ from the training distribution, the proxy yields inaccurate predictions." We will ensure to make this point clearer in our revision to make the concept and its importance more accessible.
>
> > There is no experimental evidence to support the assertion that the tri-mentoring approach can effectively address the out-of-distribution problem.
>
> Thank you for raising concerns regarding the experimental validation of our tri-mentoring approach's efficacy against out-of-distribution issues.
>
> To directly tackle this concern, we have performed additional experiments on TFBind8 and TFBind10. We choose the two tasks as we could easily identify high-scoring designs and thereby create an out-of-distribution test set. From each dataset, we sample 1000 high-scoring designs to form our OOD test sets. For these, our tri-mentoring ensemble have been tested over 5 runs against a mean ensemble and a single proxy, and the mean squared error (MSE) is used as the metric. Here are the results:
>
> - **TFBind8**:
>   - Tri-mentoring ensemble: MSE of $0.0537 \pm 0.0008$
>   - Mean ensemble: MSE of $0.0549 \pm 0.0006$
>   - Single proxy: MSE of $0.0660 \pm 0.0012$
>
> - **TFBind10**:
>   - Tri-mentoring ensemble: MSE of $0.2271 \pm 0.0075$
>   - Mean ensemble: MSE of $0.2669 \pm 0.0084$
>   - Single proxy: MSE of $0.2776 \pm 0.0185$
>
> These results illustrate that the tri-mentoring ensemble consistently outperforms the other models in handling OOD designs. We'll incorporate these experiments and analysis into Appendix.
>
> These findings, coupled with our main manuscript and appendices, offer a comprehensive demonstration of our method's efficacy. Specifically, in Line 234, we highlight, "(1) Table 1 demonstrates *tri-mentoring* attains the best results across the board, highlighting its effectiveness. Its consistent gains over Grad confirm its ability to tackle the out-of-distribution issue." Furthermore, in Appendix A.3 "Accuracy of Pairwise Consensus Labels," we illustrate that a majority of the consensus labels are accurate, underscoring our approach's competency in addressing the out-of-distribution problem.
>
> In light of this evidence, we firmly believe in the capability of our tri-mentoring approach to effectively address the out-of-distribution problem.
>
> ## Questions
>
> > How does leveraging the knowledge of the static dataset help to learn more accurate soft-labels?
>
> We greatly appreciate your question. The primary benefit of utilizing the static dataset comes from the commonality that it shares with the soft labels. Despite having different data distributions, the soft-labels and the static dataset share underlying similarities, as they both represent the same ground-truth from pairwise and pointwise perspectives, respectively.  This inherent connection allows for a rich exchange of information between the two, significantly benefiting the learning of more accurate soft-labels.
>
> More specifically, a proxy fine-tuned with soft-labels is expected to perform well on the static dataset due to this commonality. We thus leverage the regression loss of the static dataset to guide the optimization of soft-labels using bi-level optimization. This strategy is based on our novel recognition of the shared underlying truth between the static dataset and the soft-labels, making our method uniquely positioned to capitalize on it. The specific discussion about this methodology can be found in our manuscript from Line 158 to Line 168.
>
> ## Overall
>
> Have we addressed your queries satisfactorily? Your feedback is invaluable, and we're keen on further discussions. Thank you.

---

> > ### Author Response · Authors · 2023-08-17
> > **Looking forward to your feedback**
> >
> > We sincerely appreciate the time and effort you have invested in reviewing our work. We have made diligent efforts to address all the concerns you listed, including:
> >
> > - Clarified the research question and the out-of-distribution problem and emphasized the significance of both.
> > - Provided experimental evidence to demonstrate the efficacy of our tri-mentoring approach in handling out-of-distribution issues
> > - Explained how leveraging the knowledge of the static dataset aids in the learning of more accurate soft-labels
> >
> > Could you please provide further clarification if there are any aspects that remain unclear? In the past week, we have received constructive responses from the other three reviewers. We value your perspective and are eagerly looking forward to your feedback. We stand ready to make any necessary revisions to enhance our paper.

---

> > > ### Comment · Reviewer_VCj9 · 2023-08-18
> > >
> > > Thank you very much for the responses which address and answer my questions. After reading the rebuttal and reviews of other reviewers, I decide to keep my score.

---

> > > > ### Author Response · Authors · 2023-08-18
> > > > **Thanks**
> > > >
> > > > Thank you for taking the time to consider our rebuttal and the perspectives of other reviewers. We are grateful for your decision to maintain your score, and we appreciate your thoughtful evaluation of our work.

---

### Official Review · Reviewer_usfZ · 2023-07-07

**Soundness:** 4 excellent
**Presentation:** 3 good
**Contribution:** 3 good
**Rating:** 7
**Confidence:** 4

**Summary:**

This paper introduces an innovative study that revolutionizes offline model-based optimization (offline MBO) by introducing a novel ensemble method. The proposed approach addresses a critical challenge in offline MBO, namely the handling of potentially inaccurate pseudo-labels generated by proxy models, particularly in out-of-distribution scenarios. It achieves this by harnessing the power of multiple parallel proxies, utilizing voting-based supervision, and incorporating adaptive pseudo-labeling.

One of the notable strengths of this method is its intuitive ensemble framework, which effectively mitigates the inherent inaccuracies of proxy models. Despite the technical complexity and meticulous design tailored specifically for the offline MBO task, this approach demonstrates significant performance improvements across various benchmark datasets.

I recommend accepting this paper with some minor revisions.

**Strengths:**


1. This paper presents a pioneering method that effectively mitigates the issue of proxy model inaccuracy in out-of-distribution scenarios, offering a novel solution to this longstanding challenge.

2. The bi-level formulation utilized in this study demonstrates technical robustness and holds great potential for widespread application across various domains, highlighting its versatility and reliability.

3. The approach of employing a three-proxy ensemble may appear simple at first glance, but its effectiveness in improving results is remarkable, showcasing the power of this streamlined yet impactful technique.

4. The performance of the proposed method shows great promise, achieving impressive results not only at the 100th percentile but also at the 50th percentile, indicating consistent and reliable advancements across the entire distribution.

5. The ablation study presented in this paper is exemplary in its clarity and thoroughness, providing a comprehensive analysis of the individual components and their contributions to the overall method, enhancing the scientific rigor and understanding of the approach.




**Weaknesses:**

1. The analysis lacks consideration for ensemble techniques, which could have enhanced the effectiveness of the offline MBO. Ensembles, being statistical methods, offer the opportunity for deeper analysis and intuition, especially from a statistical perspective. Incorporating such analysis and intuition into the manuscript would greatly improve its quality.

2. The algorithm appears to be overly complex, introducing numerous degrees of freedom. This complexity could limit the practicality of the offline MBO approach since there is no opportunity to fine-tune hyperparameters using an oracle function. Simplifying the algorithm and reducing the number of degrees of freedom would make it more practical and applicable in real-world scenarios.

3. It would be beneficial to provide a more comprehensive discussion on ensemble techniques in similar domains, such as offline biosequential design [1,2]. Exploring the applications of ensemble methods in these domains would strengthen the manuscript's relevance and provide valuable insights for researchers in related fields.

[1] Kim, Minsu, et al. "Bootstrapped Training of Score-Conditioned Generator for Offline Design of Biological Sequences." arXiv preprint arXiv:2306.03111 (2023).

[2] Jain, Moksh, et al. "Biological sequence design with gflownets." International Conference on Machine Learning. PMLR, 2022.

**Questions:**

1. Can this method effectively solve high-dimensional biological sequence optimization problems, such as optimizing GFP and UTR sequences?

2. Could you please provide an approximate estimate of the training time required for this method? This information would be valuable for future researchers interested in implementing the approach.

3. Does the performance of the method depend on the number of proxies used?

4. What is the rationale behind using Tri-mentoring? How would leveraging bi-mentoring between the individuals involved affect the outcomes?

5. What are the expected outcomes when incorporating conservatism (e.g., COMs) or a Gaussian prior (e.g., ROMA) into the proxy model and integrating them with the tri-mentoring idea? Can these methods be orthogonal to each other?


**Limitations:**

Authors already put the limitation part in the paper which is really valuable.

---

> ### Author Rebuttal · Authors · 2023-08-06
>
> ## General Reply
> We value your constructive feedback and will incorporate it diligently in our revisions. Thank you.
> ## Weakness
> > The analysis lacks consideration for ensemble techniques
>
> We thank you for highlighting the importance of ensemble. In our manuscript, we've employed and discussed two ensemble methods: Deep Ensemble (DE, mean ensemble) and Gradient Boosting (GB, sequentially trains proxies). As noted in Line 236-238, tri-mentoring outperforms DE and GB in all four tasks, demonstrating its robustness by utilizing shared ranking supervision signals.
>
> Further, Appendix A.3 focuses on the accuracy of consensus labels generated by the three-proxy ensemble within our tri-mentoring strategy.  A majority of these labels are accurate, offering clear intuition behind the ensemble's efficacy.
>
> > The algorithm appears to be overly complex.
>
> Refer to the global response "On Additional Hyperparameters".
>
> > It would be beneficial to provide a more comprehensive discussion on ensemble techniques in offline biosequential design [1,2].
>
> We value your suggestion on delving deeper into ensemble in offline biosequential design. While our experiments already include two tasks from design-bench: TFB8 and TFB10, to design DNA, we acknowledge the significance of [1,2].
>
> Your referenced works [1,2] also use a trained proxy, which we believe can benefit from our parallel-mentoring. To underscore this, we will enhance our discussion by adding in Line 308: "Our proposed ensemble training process, with its focus on parallel-mentoring, has the potential to improve the proxy/reward training in [1, 2], thereby contributing to advancements in biological sequence design."
>
> ## Questions
>
> > Can this method effectively solve high-dimensional biological sequence optimization
>
> Indeed, our approach has been tested and proven effective on these tasks, achieving normalized scores of 0.865 (GFP) and 0.699 (UTR), which are quite competitive compared to performances reported in [3,4].
>
> In the original manuscript, we choose not to include these results as suggested by [5] which states that many methods demonstrate indistinguishable performances on GFP and UTR. However, to highlight our method's versatility, we will add in Line 252: "Our proposed method, also demonstrates its effectiveness on high-dimensional biological sequence design, achieving maximum normalized scores of 0.865 and 0.699 on GFP and UTR respectively."
>
> > Could you please provide an approximate estimate of the training time
>
> Thank you for emphasizing the significance of training time.
>
> To provide an approximate estimate:
>
> - Single proxy training on static dataset, the time (seconds)  is:
>   - TFB8: 101.1 s
>   - Ant: 33.5 s
>
> This training time is consistent across all methods that utilize a proxy.
>
> As for tri-mentoring, the added overhead primarily stems from proxy fine-tuning. Here's the breakdown:
>   - TFB8: fine-tuning step is 0.21 s.
>   - Ant: fine-tuning step is 0.08 s.
>
> These efficient timings, coupled with our method's benefits, make our approach a practical choice for future researchers. We'll incorporate these analyses into Appendix.
>
> > Does the performance of the method depend on the number of proxies used?
>
> Refer to the global response "Regarding Number of Parallel Proxies".
>
> > What is the rationale behind using Tri-mentoring?
>
> The underpinning idea behind tri-mentoring is to harness the voted ranking signal to fine-tune the proxy. This is depicted in Figure 1 of our paper. Here, we maintain three parallel proxies, namely fA, fB, and fC. Suppose we have two designs x1 and x2 near the current point. If proxies fA and fB concur that the score of x1 is larger than that of x2, while fC disagrees, we follow the majority voting principle. The agreement between fA and fB provides a more trustworthy ranking. Their voted ranking signal fV(x1) > fV(x2) is then used to mentor fC, thereby enhancing its performance.
>
> As for bi-mentoring, it's not applicable since the majority voting we leverage necessitates at least three proxies to cast a vote, making it unfit for a bi-mentoring approach.
>
> > What are the expected outcomes when incorporating conservatism (e.g., COMs) or a Gaussian prior (e.g., ROMA)
>
> Thank you for the insightful inquiry on integrating conservatism (COMs) and Gaussian priors (ROMA) with our tri-mentoring.
>
> In brief, our additional experimental results on Ant and TFB8 tasks are:
>
> | Method | Ant | TFB8 |
> |--------|------|-----|
> | COMs | 0.856  | 0.496 |
> | Tri-mentoring | 0.948  | 0.970 |
> | COMs + Tri-mentoring | 0.941 | 0.960  |
> | ROMA | 0.914  | 0.917 |
> | Tri-mentoring | 0.948  | 0.970 |
> | ROMA + Tri-mentoring | 0.945  | 0.971 |
>
> While COMs combined with tri-mentoring leads to a slight drop in performance, the ROMA and tri-mentoring combination retains similar outcomes. The dip with COMs might arise from its lower scoring for out-of-distribution designs, potentially conflicting with the tri-mentoring's majority voting mechanism for ood designs.
>
> On the other hand, ROMA, promotes smooth proxy models. Theoretically, this characteristic should be orthogonal to our tri-mentoring. The absence of improvement might suggest that our current benchmark is not sufficiently challenging to exploit the combined strengths of both ROMA and tri-mentoring. We'll ensure to incorporate these analyses into Appendix.
>
> ## Overall
> Does our response address your concerns? We look forward to continued discussion. Thank you.
>
>     [1] Kim, Minsu, et al. Bootstrapped Training of Score-Conditioned Generator for Offline Design of Biological Sequences.
>     [2] Jain, Moksh, et al. Biological sequence design with gflownets.
>     [3] Trabucco B, Kumar A, Geng X, et al. Conservative objective models for effective offline model-based optimization.
>     [4] Chen C, Zhang Y, Fu J, et al. Bidirectional learning for offline infinite-width model-based optimization.
>     [5] Trabucco B, Geng X, Kumar A, et al. Design-bench: Benchmarks for data-driven offline model-based optimization.

---

> > ### Comment · Reviewer_usfZ · 2023-08-17
> >
> > Thank you for the rebuttal. I keep my score supporting this paper to be accepted.

---

> > > ### Author Response · Authors · 2023-08-17
> > > **Thanks for your continued support.**
> > >
> > > Thank you for your continued support and for maintaining your score in favor of accepting our paper. We truly appreciate your time and the insights you provided throughout the review process.

---

### Official Review · Reviewer_CizP · 2023-07-16

**Soundness:** 2 fair
**Presentation:** 3 good
**Contribution:** 2 fair
**Rating:** 5
**Confidence:** 4

**Summary:**

The paper studies offline model-based optimization, which maximizes a black-box objective with a dataset of designs and scores. Most approaches rely on training a dataset proxy to approximate the black-box objective and perform SGD on the objective. This paper proposes using three proxies and voting-based supervision to generate labels. In addition, they use adaptive soft-labeling to make the proposed method more robust to label noises.

**Strengths:**

+ A clear description of the problem background and the proposed algorithm procedure;
+ Extensive ablation studies on various components, such as the voting based pairwise supervision;


**Weaknesses:**

- It's weak and less general that the proposed method constrains itself into the three parallel mentoring cases. I'd suggest the writing changed to a much more general form of any number of the parallel proxies;

- The adaptive soft labeling seems to directly apply the meta-learning method /bi-level optimization method on the pseudo-label/soft label. The novelty seems not very justifiable.

**Questions:**

Please address the two weaknesses above.

---

> ### Author Rebuttal · Authors · 2023-08-06
>
> ## General Reply
>
> We sincerely appreciate the effort and time you've invested in providing us with your insightful and constructive feedback. Your comments are invaluable to us as they offer opportunities for refining, rectifying, and enhancing the content of this paper. We are committed to meticulously revising our paper based on the suggestions you've delineated below.
>
> ## Weaknesses
>
> > It's weak and less general that the proposed method constrains itself into the three parallel mentoring cases. I'd suggest the writing changed to a much more general form of any number of the parallel proxies;
>
>
> We appreciate your feedback concerning the perceived limitation of our method to three parallel mentoring instances. However, it's important to clarify that we **have indeed considered a more general framework in our paper**, extending beyond the three-proxy scenario.
>
> As stated in the paper's appendix under the subsection titled "Scenarios for Parallel Mentoring with Multiple Proxies", we have extended our method to include multiple proxies. The performance is quite robust against the number of parallel proxies.
>
> Furthermore, we **explicitly reference this aspect in three instances within the main body of our paper**:
>
> 1. Lines 57-58: "This paper primarily focuses on the three-proxy case, referred to as tri-mentoring, but we also examine the situation with more proxies in Appendix A.1."
>
> 2. Lines 106-107: "The method can be easily extended to incorporate more proxies, as discussed in
> 107 Appendix A.1."
>
> 3. Lines 174-175: "While our primary focus is the tri-mentoring with 3 proxies, we additionally explore other parallel-mentoring implementations utilizing 5, 7, 9, and 11 proxies in Appendix A.1."
>
> Therefore, we believe that we have thoroughly addressed your concerns about the general applicability of our method, as described above. We also encourage you to refer to the global response "Regarding Number of Parallel Proxies".
>
> > The adaptive soft labeling seems to directly apply the meta-learning method /bi-level optimization method on the pseudo-label/soft label. The novelty seems not very justifiable.
>
> We appreciate your thoughtful comment regarding the perceived similarity between our proposed adaptive soft labeling approach and existing bi-level optimization methods applied to soft labels[1]. However, our approach carries substantial novelty due to its fundamental differences from previous works [1] that also applied bi-level optimization to optimize soft labels. These differences lie in three key aspects: initialization, optimization, and objective.
>
> 1. **Initization**: Contrary to previous works that aim to address label noise in a collected dataset, our method handles label noise created by a unique voting mechanism along the gradient optimization path.
>
> 2. **Optimization**: Previous work typically leverages the classification loss of a clean validation set to update the soft label. However, our method proposes an innovative strategy of employing the regression loss of the static dataset for soft label updates. This unique strategy is due to our novel recognition that, despite their differing data distributions, the soft-labels and the static dataset share significant underlying similarities. They both represent the same ground-truth from pairwise and pointwise perspectives, respectively.
>
> 3. **Objective**: Whereas previous works primarily focus on mitigating label noise to enhance the performance of a classification model, our work takes a distinctive route by addressing label noise to improve the performance of a regression model for offline MBO. This adaptation to a new context further underscores the innovative aspects of our approach.
>
> We will incorporate these discussions into Appendix.
>
> ## Overall
> Does the provided response address your concerns? We greatly value your thorough feedback and anticipate further discussion during the rebuttal phase. Thank you.
>
>     [1] Wu Y, Shu J, Xie Q, et al. Learning to purify noisy labels via meta soft label corrector. AAAI 2021.

---

> > ### Comment · Reviewer_CizP · 2023-08-17
> >
> > Thanks for pointing out that the more general formulation has been discussed in the Appendix, which has addressed my major concerns.

---

> > > ### Author Response · Authors · 2023-08-17
> > > **Thanks for your support.**
> > >
> > > Thank you for your adjusted rating of 5 and for recommending acceptance. We are pleased to hear that our response in the Appendix has addressed most of your major concerns. We appreciate the time and effort you've put into reviewing our work.

---

### Author Rebuttal · Authors · 2023-08-06

Dear Reviewers,

Thank you for your thorough examination of our paper and for sharing insightful feedback. We recognize your concerns and, in this rebuttal, address two main common points raised.

## On Additional Hyperparameters

> review from usfZ: The algorithm appears to be overly complex, introducing numerous degrees of freedom. This complexity could limit the practicality of the offline MBO approach since there is no opportunity to fine-tune hyperparameters using an oracle function. Simplifying the algorithm and reducing the number of degrees of freedom would make it more practical and applicable in real-world scenarios.

> review from NEb9: My primary concern revolves around hyperparameter selection. It appears the method introduces a considerable number of additional hyperparameters, including, but not limited to, learning rates for fine-tuning soft-labels and proxy models, the variance
 for selecting an adjacent input from the current solution, and the number of near inputs,. Although the authors demonstrated the stability and effect of these hyperparameters (e.g., ), an analysis of some other hyperparameters, such as or fine-tuning learning rates, is still missing. Given that the paper addresses "offline" MBO, the experiments should detail how these hyperparameters can be determined (e.g.,
 might be defined as a function of input dimension or a constant). Alternatively, the authors should demonstrate that the proposed method is robust to variations in these hyperparameters.


We understand the concerns related to the algorithm's complexity and our focus has been on demonstrating its robustness with different hyperparameters. We have four extra introduced parameters: 1. number of samples ($K$), 2. variance $\delta$, 3. fine-tuning learning rate ($\gamma$) and 4. soft-labeling learning rate ($\lambda$). As detailed in Section 4.6 and Appendix A.4, our method remains **robust against the number of Samples ($K$) and  variance $\delta$**. As for $\gamma$ and $\lambda$, while we've set default constants (1e-3 for $\gamma$ and 1e-1 for $\lambda$), we've also conducted additional experiments over varied ranges on TFB8 and Ant:

   - Evaluated across:
     - $\gamma$: [2.5e-4, 5e-4, 1e-3, 2e-3, 4e-3]
     - $\lambda$: [0.025, 0.05, 0.1, 0.2, 0.4]

The following results are normalized by dividing them by the result obtained for the default values.

Experimental Results on TFB8
| Parameter | 2.5e-4 | 5e-4 | 1e-3 | 2e-3 | 4e-3 |
|----------|-------|------|------|------|------|
| $\gamma$ | 0.903 | 0.995 | 1.000 | 0.996 | 1.006 |
| $\lambda$ | 0.945 | 0.983 | 1.000 | 1.000 | 0.995 |

Experimental Results on Ant
| Parameter | 2.5e-4 | 5e-4 | 1e-3 | 2e-3 | 4e-3 |
|----------|-------|------|------|------|------|
| $\gamma$ | 0.953 | 1.008 | 1.000 | 1.021 | 1.022 |
| $\lambda$ | 1.011 | 0.989 | 1.000 | 0.992 | 0.997 |

These results confirm the method's **resilience against hyperparameter variations on $\gamma$ and $\lambda$**, emphasizing its potential for real-world scenarios despite its perceived complexity. We'll ensure to incorporate these analyses into Appendix A.4 for better clarity.

## Regarding Number of Parallel Proxies

> review from CizP: It's weak and less general that the proposed method constrains itself into the three parallel mentoring cases. I'd suggest the writing changed to a much more general form of any number of the parallel proxies.

> review from usfZ: Does the performance of the method depend on the number of proxies used?


Our paper indeed places an emphasis on the three-proxy configuration, commonly referred to as "tri-mentoring". However, this focus does not imply a limitation. We have actively pursued a broader framework, one that encompasses multiple proxies.

As stated in the paper's appendix under the subsection titled "Scenarios for Parallel Mentoring with Multiple Proxies", we **have extended our method to include multiple proxies**. Performance remains robust across different numbers of parallel proxies. As the number of proxies (M) increases, the performance ratios for both tasks generally improve, eventually reaching a plateau. This behavior suggests that an increased number of proxies enhances the robustness of the ensemble due to the increased diversity.

Furthermore, we **have explicitly referenced this aspect** in three instances within the main body of our paper:

1. Lines 57-58: "This paper primarily focuses on the three-proxy case, referred to as tri-mentoring, but we also examine the situation with more proxies in Appendix A.1."

2. Lines 106-107: "The method can be easily extended to incorporate more proxies, as discussed in
107 Appendix A.1."

3. Lines 174-175: "While our primary focus is the tri-mentoring with 3 proxies, we additionally explore other parallel-mentoring implementations utilizing 5, 7, 9, and 11 proxies in Appendix A.1."

Best,

Submission 51 Authors.

---

### Decision · Program_Chairs · 2023-09-21

**Decision:**

Accept (poster)

**Comment:**

This submission introduced an interesting model-based optimization method. The novel component is coming from utilizing weak ranking supervision signals among proxies with a voting strategy. The submission suggests a novel method and also provides an extensive evaluation results. All reviewers consistently gave positive reviews with similar thoughts. We believe it is a good submission to be shared at the conference. The only major feedback is to simplify the method description to ease the unncessary complicacy.